# MULTI-PLAY MULTI-ARMED BANDITS WITH SCARCE SHAREABLE ARM CAPACITIES

## ABSTRACT

This paper revisits multi-play multi-armed bandit with shareable arm capacities problem (MP-MAB-SAC), for the purpose of revealing fundamental insights on the statistical limits and data efficient learning. The MP-MAB-SAC is tailored for resource allocation problems arsing from LLM inference serving, edge intelligence, etc. It consists of $K$ arms and each arm $k$ is associated with an unknown but deterministic capacity $m_k$ and per-unit capacity reward with mean $\mu_k$ and $\sigma$ sub-Gaussian noise. The aggregate reward mean of an arm scales linearly with the number of plays assigned to it until the number of plays hit the capacity limit $m_k$, and then the aggregate reward mean is fixed to $m_k\mu_k$. At each round only the aggregate reward is revealed to the learner. Our contributions are three folds. 1) *Sample complexity:* we prove a minmax lower bound for the sample complexity of learning the arm capacity $\Omega(\frac{\sigma^2}{\mu_k^2}\log\delta^{-1})$, and propose an algorithm to exactly match this lower bound. This result closes the sample complexity gap of Wang et al. (2022a), whose lower and upper bounds are $\Omega(\log\delta^{-1})$ and $O(\frac{m_k^2\sigma^2}{\mu_k^2}\log\delta^{-1})$ respectively. 2) *Regret lower bounds:* we prove an instance-independent regret lower bound $\Omega(\sigma\sqrt{TK})$ and instance-dependent regret lower bound $\Omega(\sum_{k=1}^{K}\frac{c\sigma^2}{\mu_k^2}\log T)$. This result provides the first instance-independent regret lower bound and strengths the instance-dependent regret lower bound of Wang et al. (2022a) $\Omega(\sum_{k=1}^{K}\log T)$. 3) *Data efficient exploration:* we propose an algorithm named `PC-CapUL`, in which we use prioritized coordination of arm capacities upper/lower confidence bound (UCB/LCB) to efficiently balance the exploration vs. exploitation trade-off. We prove both instance-dependent and instance-independent upper bounds for `PC-CapUL`, which match the lower bounds up to some acceptable model-dependent factors. This result provides the first instance-independent upper bound, and has the same dependence on $m_k$ and $\mu_k$ as Wang et al. (2022a) with respect to instance-dependent upper bound. But there is less information about arm capacity in our aggregate reward setting. Numerical experiments validate the data efficiency of `PC-CapUL`.

## 1 INTRODUCTION

Multi-play multi-armed bandit (MP-MAB) is a natural and popular variant of the vanilla multi-armed bandits framework Anantharam et al. (1987a). MP-MAB has various applications such as online advertising Lagrée et al. (2016); Komiyama et al. (2017); Yuan et al. (2023), power system Lesage-Landry & Taylor (2017), mobile edge computing Chen & Xie (2022); Wang et al. (2022a); Xu et al. (2023), etc. The canonical MP-MAB model consists of a number $K \in \mathbb{N}_+$ arms. Each round the learner assigns $K$ plays to arms, where each arm can be pulled by at most one play. Once an arm is pulled, a reward is generated, which is modeled as a sample from a random variable with unknown mean and known tail property such as standard sub-Gaussian tail. The research line of MP-MAB is still active, evidenced by various recent generalizations of MP-MAB Chen & Xie (2022); Moulos (2020); Xu et al. (2023); Wang et al. (2022a); Yuan et al. (2023).

One notable generalization of MP-MAB is MP-MAB-SAC, which enables each arm with a finite number of shareable capacities Xu et al. (2023); Wang et al. (2022a). The key idea is modeling each arm with a finite capacity and allowing multiple plays to be assigned to the same arm. This

generalization provides a finer capturing of the resource sharing nature of resource allocation problems arising from LLM inference serving, edge intelligence, etc. Formally, Xu et al. (2023); Wang et al. (2022a)'s model considers a finite number of $K \in \mathbb{N}_+$ arms and a finite number of $N \in \mathbb{N}_+$ plays. Each arm $k$ is characterized by a tuple $(m_k, \mu_k, \sigma)$, where $m_k \in \mathbb{N}_+$ models the capacity limit and $\mu_k \in \mathbb{R}_+$ models the unit-capacity reward mean. Both $m_k$ and $\mu_k$ are unknown to the learner and the arm capacity $m_k$ is deterministic. The reward function of assigning $a_k \in \mathbb{N}_+$ to arm $k$ is modeled as:

$$\text{Wang et al. (2022a)'s Reward Model}: R_k(a_k) = \min\{a_k, m_k\}(\mu_k + \epsilon_k), \tag{1}$$

where $\epsilon_k$ is a zero mean $\sigma$ sub-Gaussian random noise. Wang et al. (2022a)'s main results can be summarized as:

$$\text{Sample complexity}: \Omega(\log \delta^{-1})(\text{lower bound}), \ O\left(\frac{\sigma^2 m_k^2}{\mu_k^2} \log \delta^{-1}\right)(\text{upper bound}), \tag{2}$$

$$\text{Regret lower bound}: \Omega\left(\sum_k \log T\right)(\text{rough bound, instance-dependent}), \tag{3}$$

$$\text{Regret upper bound}: O\left(\sum_k \frac{w_k \sigma^2 m_k^2}{\mu_k^2} \log T\right)(\text{rough bound, instance-dependent}). \tag{4}$$

In fact, the sample complexity lower bound and regret lower bound stated in Wang et al. (2022a) are $\Omega\left((\sigma^2 m_k^2/\mu_k^2) \log \delta^{-1}\right)$ and $\Omega((\sum_k \sigma^2 m_k^2/\mu_k^2) \log T)$ respectively. However these two bounds hold under the same condition $\mu_k^2/(\sigma^2 m_k^2) \geq 2$ (Theorem 4.1 and Theorem 4.3 of Wang et al. (2022a)), which implies that $(\sigma^2 m_k^2)/\mu_k^2 \leq 0.5$, yielding the sample complexity lower bound $\Omega(\log \delta^{-1})$ and regret lower bound $\Omega\left(\sum_k \log T\right)$.

Note that (2) implies a large sample complexity gap, while 3 and 4 implies a large regret gap. Motivated by narrowing these gaps, we revisit the MP-MAB-SAC problem, aiming to reveal fundamental insights on statistical limits and data efficient learning. Note that the reward function (1), encodes the capacity in both the mean $\mathbb{E}[R_k(a_k)] = \min\{a_k, m_k\}\mu_k$. and variance $\text{Var}[R_k(a_k)] = (\min\{a_k, m_k\})^2 \text{Var}[\epsilon_k]$. To understand the essentials, first we reduce the capacity information in the reward to the minimum such that only the reward mean contains the capacity information. Formally, we propose a new reward function to achieve this goal:

$$R_k(a_k) = \min\{a_k, m_k\}\mu_k + \epsilon_k. \tag{5}$$

Note that 5 finds its root in the reward model of conventional linear bandits with one dimensional feature Lattimore & Szepesvári (2020). One can check that under (5), only the reward mean encodes the arm capacity. Intuitively, the learning of the arm capacity would be harder than (1), and the insights derived from (5) should be more fundamental. Wang et al. (2022a) considered the capacity-abundant setting with $N < M$, where $M := \sum_{k=1}^K m_k$, which is not suitable enough for real-world severe competition under scarce resources. We thus focus on the capacity scarce setting with $N \geq M$, for the purpose of understanding the exploration vs. exploitation trade-off under severe capacity constraint. Assigning a play to an arm generates a constant movement cost $c \in \mathbb{R}_+$, which is assumed to satisfy $c < \min_k \mu_k$ and adds a cost constraint for exploration.

**Applications of MP-MAB-SAC.** MP-MAB-SAC is a versatile model with multiple applications in real world. It is illustrated in Wang et al. (2022a) that MP-MAB-SAC can be applied to edge computing, cognitive ratio applications , online advertisement placement etc. To avoid repetitive narration, we will provide another instance of MP-MAB-SAC application. Here we elaborate on how to map our model to LLM inference serving applications Li et al. (2024). Each arm model can be mapped as a deployment instance of an LLM. Arm capacity models the number of queries that an LLM can process at a given time slot. Due to multiplexing behavior of computing systems, the capacity is unknown and the processing is uncertain Zhu et al. (2023). An LLM deployed on more powerful computing facilities would be modeled with larger capacity. The reward mean $\mu_k$ can be mapped as the capability of an LLM such as large, medium and small LLM mixed inference serving. The cost $c$ can be mapped as the communication cost generated by transmitting queries to the commercial LLM server.

## 1.1 MAIN RESULTS AND CONTRIBUTIONS

Contributions of this paper can be summarized into the following three folds.

**Sample complexity.** We prove a minmax lower bound for the sample complexity of learning the arm capacity $\Omega(\frac{\sigma^2}{\mu_k^2} \log \delta^{-1})$, and propose an active inference algorithm named `ActInfCap` to exactly match this lower bound. This result closes the sample complexity gap of Wang et al. (2022a), whose lower and upper bounds are $\Omega(\log \delta^{-1})$ and $O(\frac{m_k^2 \sigma^2}{\mu_k^2} \log \delta^{-1})$ respectively. The new finding here is that the difficulty of learning the arm capacity is determined by the per-capacity reward mean. `ActInfCap` contributes new uniform confidence intervals for the arm capacity estimation and new idea of actively probing an arm with its capacity's UCB or LCB for data efficient learning of arm capacity. And the UCB or LCB are adopted alternatively in the data gathering process. These findings shed new lights on arm capacity estimation and serving building blocks for designing data efficient exploration algorithms.

**Regret lower bounds.** We prove an instance-independent regret lower bound $\Omega(\sigma\sqrt{TK})$ and instance-dependent regret lower bound $\Omega(\sum_{k=1}^{K} \frac{c\sigma^2}{\mu_k^2} \log T)$. This result provides the first instance-independent regret lower bound and strengths the instance-dependent regret lower bound of Wang et al. (2022a) $\Omega(\sum_{k=1}^{K} \log T)$. Our regret lower bounds have no dependence on the arm capacity $m_k$. At the first glance, this looks counterintuitive, however it is aligned with our sample complexity lower bound which states that the sample complexity is independent of the arm capacity. Also the dependence on the reward mean is aligned with the sample complexity. The finding here is that the difficulty of learning the optimal action is basically limited by the number of arms $K$ and the per-unit capacity reward mean $\mu_k$. Increasing the number of arms or decreasing the reward mean would make the learning more difficult.

**Data efficient exploration.** We propose an algorithm named `PC-CapUL`, in which we use prioritized coordination of arm capacities upper/lower confidence bound (UCB/LCB) to efficiently balance the exploration vs. exploitation trade-off. We prove both instance-dependent and instance-independent upper bounds for `PC-CapUL`, which match the lower bounds up to some acceptable model-dependent factors. These results provide the first instance-independent upper bound, and have the same dependence on $m_k$ and $\mu_k$ as Wang et al. (2022a) in respect of the instance-dependent upper bound. But there is less information about arm capacity in our aggregate reward setting. Numerical experiments validate the data efficiency of `PC-CapUL`. The main idea of `PC-CapUL` has four folds: (1) *Preventing excessive UEs.* At each time slot, ensure that the number of individual exploration (IE), is no less than the number of united exploration (UE), where UE/IE means that the number of plays assigned to an arm equals its capacities' UCB/LCB. (2) *Balancing UE and IE.* At each time slot, let as many arms as possible to do UEs, inspired by the insight from Lemma 5 revealing that both UE and IE are required to reach their corresponding limits. (3) *Favorable arms win UE first.* At each time slot, in cases when multiple arms compete for UEs, we resolve this competition via larger-empirical-reward-mean-first rule. The insight is that it is easier to learn the capacity $m_k$ if the unit utility $\mu_k$ is larger. (4) *Stop learning when converges.* At each time slot, once an arm's capacity upper bound and lower bound meet with each other, there should be no more exploration on that arm.

## 2 RELATED WORK

To the best of our knowledge, MP-MAB was first studied by Anantharam *et al.* Anantharam et al. (1987a), where an asymptotic regret lower bound was established and an algorithm achieving the lower bound asymptotically was proposed. The regret lower bound in the finite time is achieved by *et al.* Komiyama et al. (2015) via Thompson sampling. Markovian rewards variant of MP-MAB wa studied in Anantharam et al. (1987b). Some recent generalization of MP-MAB include: cascading MP-MAB where the order of plays is captured into the reward function Lagrée et al. (2016); Komiyama et al. (2017), MP-MAB with switching cost Agrawal et al. (1990); Jun (2004), MP-MAB with budget constraint Luedtke et al. (2019); Xia et al. (2016); Zhou & Tomlin (2018) and MP-MAB with a stochastic number of plays in each round Lesage-Landry & Taylor (2017), sleeping MP-MAB *et al.* Yuan et al. (2023), MP-MAB with shareable arm capacities Chen & Xie (2022); Wang et al. (2022a); Xu et al. (2023).

Our work falls into the research line of MP-MAB with shareable arm capacities Chen & Xie (2022); Wang et al. (2022a;b); Xu et al. (2023); Mo & Xie (2023). The shareable arm capacities models can be categorized into two types: (1) stochastic arm capacity but with feedback on the realization of arm capacity Chen & Xie (2022); Mo & Xie (2023); (2) deterministic capacity without any realization of the arm capacity Wang et al. (2022a;b); Xu et al. (2023). Though the difference looks small, the two settings lead to fundamentally different research problems and techniques for address it. For the stochastic arm capacity line, Chen *et al.* Chen & Xie (2022) models the arm capacity as a random variable, but in each round the sample of the arm capacity of all arms are revealed to the decision, i.e., expert feedback on arm capacity. One can directly estimate the distribution of arm capacity from the capacity samples. Mo & Xie (2023) generalizes this model to the distributed setting, and uses the realization of the arm capacity as a signal for coordination. However, the deterministic arm capacity is technically different. Though the capacity is deterministic, it is unknown and on the decision maker can only access samples from the reward function, while no samples on the arm capacity can be observed. Wang et al. (2022a;b); Xu et al. (2023). Xu et al. (2023) consider the setting in which multiple strategic agents compete for the resource. Nash equilibrium in the offline setting is established. Our work revisits this research line. Our work is motivated by the observation that the condition $\mu_k^2/\sigma_k^2 m_k^2 \geq 2$ that guarantees the sample complexity lower bound and regret lower bound of Wang et al. (2022a) implies that theses two bounds reduces to $\Omega(\log \delta^{-1})$ and $\Omega(\sum_k \log T)$, namely trivial lower bound. This implies a huge gap between the upper and lower bound. We thus revisit this problem, aiming for a deeper understanding of this problem. We close the sample complexity gap and narrow the regret gap (please refer to introduction for details).

## 3 MODEL & PROBLEM FORMULATION

**Notation:** By default, for any integer $N \in \mathbb{N}_+$: $[N] := \{1, \ldots, N\}$.

Consider $K \in \mathbb{N}_+$ arms indexed by $[K]$ and $N \in \mathbb{N}_+$ plays to be assigned to these arms. Each arm $k \in [K]$ is characterized by a tuple $(m_k, \mu_k, \sigma)$, where $m_k \in [N]$ and $\mu_k \in \mathbb{R}$ and $\sigma \in \mathbb{R}$. Here, $m_k$ models the capacity of arm $k$, $\mu_k$ models the per-unit reward mean of arm $k$, and $\sigma \in \mathbb{R}_+$ models tail property of the reward, i.e., $\sigma$ sub-Gaussian. Both $m_k$ and $\mu_k$ are unknown to the learner, and the capacity $m_k$ is deterministic. We consider the scarce arm capacity setting, such that $N \geq M$, where $M := \sum_{k=1}^{K} m_k$ denotes the total amount of capacities across all arms. For every play there is a constant movement cost $c$ to an arm, which is known to the learner. The movement cost can model the charge of each query in LLM inference serving applications, the transmission cost in edge intelligence application, etc. From a learning perspective, it adds a cost constraint to exploration. Let $a_k \in [N]$ denotes the number of plays assigned to arm $k \in [K]$. The reward function associated with $a_k$ is stated in (5).

Consider $T \in \mathbb{N}_+$ time slots. Let $a_{k,t} \in [N] \cup \{0\}$ denote the number of plays assigned to the arm $k$ at time slot $t$, and the action made in the slot $t$ is characterized by the vector $\mathbf{a}_t := (a_{1,t}, a_{2,t}, ..., a_{K,t})$. The action space $\mathcal{A}$ is:

$$\mathcal{A} := \left\{ (a_1, a_2, ..., a_K) \in \mathbb{N}^K \middle| \sum_{k \in [K]} a_k \leq N \right\}.$$

Denote the utility of the action $\mathbf{a}_t$ at time slot $t$ on arm $k$ as $U_{k,t}$, which is defiend as the reward minus movement cost:

$$U_{k,t}(a_{k,t}) := R_k(a_{k,t}) - c \cdot a_{k,t}.$$

We then define the expected utility for action $\mathbf{a}_t$ as $f(\mathbf{a})$:

$$f(\mathbf{a}) := \mathbb{E}\left[ \sum_{k \in [K]} U_k(a_k) \right] = \sum_{k \in [K]} (\min\{a_k, m_k\} \cdot \mu_k - c \cdot a_k)$$

Let $\mathbf{a}^*$ denote the optimal action $\mathbf{a}$ that maximizes the expected utility $f(\mathbf{a})$,i.e.:

$$\mathbf{a}^* := \arg\max_{\mathbf{a}} f(\mathbf{a})$$

And it is obvious that the optimal action is $\mathbf{a}^* = (m_1, m_2, ..., m_K)$. The difficulty then lies on how to distinguish the capacities of all the arms and the order is important in this problem. The objective

is to minimize the regret over $T$ time slots, which is defined as $\text{Reg}_T(T)$:

$$\text{Reg}_T(T) := \mathbb{E}\left[Tf(\mathbf{a}^*) - \sum_{t=1}^{T} f(\mathbf{a}_t)\right].$$

## 4 SAMPLE COMPLEXITY OF ESTIMATING ARM CAPACITY

### 4.1 SAMPLE COMPLEXITY LOWER BOUND

We focus on understanding the hardness of inferring the arm capacities, since this determines the optimal allocation of plays. We consider the setting that given a fixed arm $k$, an inference algorithm $\pi_{\text{Inf}}$ generates samples by assigning $a_{k,t} \in [N]$ plays to it.

**Definition 1** (Wang et al. (2022a)). *An action $a_{k,t}$ is United Exploration (UE) if $a_{k,t} > m_k$. An action $a_{k,t}$ is individual exploration (IE) if $a_{k,t} \leq m_k$.*

Note that $1 \leq m_k < N$ is taken as a prior, so both UE and IE are possible for $\pi_{\text{Inf}}$. We consider a space of all the inference algorithm $\pi_{\text{Inf}}$ that can adaptively vary the numbers of UE and IE.

**Theorem 1.** *For any inference algorithm $\pi_{\text{Inf}}$, there exists an instance of arm $k$ such that:*

$$\mathbb{P}\left[\hat{m}_{k,t} \neq m_k \middle| t \leq \frac{2\sigma^2}{\mu_k^2}\log\left(\frac{1}{4\delta}\right)\right] \geq 1 - \delta,$$

*where $\hat{m}_{k,t}$ denotes the estimator of arm capacity produced by $\pi_{\text{Inf}}$.*

**Remark.** Theorem 1 establishes a minmax lower bound $\Omega(\frac{\log \delta^{-1}}{\mu_k^2})$ for the sample complexity of estimating arm capacity. It significantly strengths the lower bound $\Omega(\log \delta^{-1})$ of Wang et al. (2022a). The new finding here is that the difficulty of learning the arm capacity is determined by the per-capacity reward mean and it is independent of the arm capacity $m_k$. This theorem is proved by applying the Le Cam's method with a careful tracking of the number of UEs.

### 4.2 SAMPLE EFFICIENT ALGORITHM

**Uniform confidence interval for arm capacity.** First we formally define $\tau_{k,t}$ and $\iota_{k,t}$ as the number of IE and UE on arm $k$ up to time slot $t$:

$$\tau_{k,t} = \sum_{s=1}^{t} \mathbb{1}\{a_{k,s} \leq m_k\}, \quad \iota_{k,t} = \sum_{s=1}^{t} \mathbb{1}\{a_{k,s} > m_k\}$$

And since in training process the real capacity $m_k$ is unknown, we should use the confidence interval rather than the capacity itself to calculate an empirical version of $\tau_{k,t}$ and $\iota_{k,t}$. Then we define the empirical version of $\tau_{k,t}$ and $\iota_{k,t}$ as $\hat{\tau}_{k,t}$ and $\hat{\iota}_{k,t}$:

$$\hat{\tau}_{k,t} = \sum_{s=1}^{t} \mathbb{1}\{a_{k,s} \leq m_{k,s-1}^l\}, \quad \hat{\iota}_{k,t} = \sum_{s=1}^{t} \mathbb{1}\{a_{k,s} \geq m_{k,s-1}^u\}$$

Another term we need is the scaling factor of IE:

$$\psi_{k,t} = \frac{1}{\tau_{k,t}}\sum_{s=1}^{t} a_{k,s}\mathbb{1}\{a_{k,s} \leq m_k\}, \quad \hat{\psi}_{k,t} = \frac{1}{\hat{\tau}_{k,t}}\sum_{s=1}^{t} a_{k,s}\mathbb{1}\{a_{k,s} \leq m_{k,s-1}^l\}$$

The estimator of $\mu_k$ up to time slot $t$ is defined as $\hat{\mu}_{k,t}$. Let $\upsilon_k := m_k\mu_k$ and the estimator of $m_k\mu_k$ up to time slot $t$ is defined as $\hat{\upsilon}_{k,t}$:

$$\hat{\mu}_{k,t} = \left(\sum_{s=1}^{t}\left(U_{k,s}(a_{k,s}) + c \cdot a_{k,s}\right)\mathbb{1}\left\{a_{k,s} \leq m_{k,s-1}^l\right\}\right)\Big/(\hat{\tau}_{k,t}\hat{\psi}_{k,t}), \tag{6}$$

$$\hat{\upsilon}_{k,t} = \left(\sum_{s=1}^{t}\left(U_{k,s}(a_{k,s}) + c \cdot a_{k,s}\right)\mathbb{1}\left\{a_{k,s} \geq m_{k,s-1}^u\right\}\right)\Big/\hat{\iota}_{k,t}. \tag{7}$$

To simplify notation, we denote the function :

$$\phi(x,\delta) := \sqrt{\left(1 + \frac{1}{x}\right)\frac{2\log\left(2\sqrt{x+1}/\delta\right)}{x}}.$$

**Lemma 1.** *Then the confidence intervals of the estimator $\hat{\mu}_{k,t}$ and $\hat{\upsilon}_{k,t}$ can be calculated as:*

$$\hat{\mu}_{k,t} \in \left[ \mu_k - \sigma\phi\left(\hat{\tau}_{k,t}, \delta\right)/\hat{\psi}_{k,t}, \mu_k + \sigma\phi\left(\hat{\tau}_{k,t}, \delta\right)/\hat{\psi}_{k,t} \right] \tag{8}$$

$$\hat{\upsilon}_{k,t} \in \left[ \upsilon_k - \sigma\phi\left(\hat{\iota}_{k,t}, \delta\right), \upsilon_k + \sigma\phi\left(\hat{\iota}_{k,t}, \delta\right) \right] \tag{9}$$

*For fixed $k$, these confidence intervals are correct for all $t \in [T]$ with probability at least $1 - \delta$.*

Noticing that $\upsilon_k = m_k \mu_k$, we rearrange the terms in the confidence interval (8) (9) and get:

$$\mu_{k,t} \in \left[ \hat{\mu}_{k,t} - \sigma\phi\left(\hat{\tau}_{k,t}, \delta\right)/\hat{\psi}_{k,t}, \hat{\mu}_{k,t} + \sigma\phi\left(\hat{\tau}_{k,t}, \delta\right)/\hat{\psi}_{k,t} \right]$$

$$m_k \mu_k \in \left[ \hat{\upsilon}_{k,t} - \sigma\phi\left(\hat{\iota}_{k,t}, \delta\right), \hat{\upsilon}_{k,t} + \sigma\phi\left(\hat{\iota}_{k,t}, \delta\right) \right]$$

Use the endpoints of the interval above and then we can get the lemma about the arm capacity confidence interval.

**Lemma 2.** *For any adaptive algorithm thus uses first $K$ time slots for initialization. If $\sigma\phi\left(\hat{\tau}_{k,t}, \delta\right)/\hat{\psi}_{k,t} < \hat{\mu}_{k,t}$, the event $A_k$:*

$$A_k := \left\{ \forall t \in [T], t > K, m_k \in \left[ \frac{\hat{\upsilon}_{k,t} - \sigma\phi\left(\hat{\iota}_{k,t}, \delta\right)}{\hat{\mu}_{k,t} + \sigma\phi\left(\hat{\tau}_{k,t}, \delta\right)/\hat{\psi}_{k,t}}, \frac{\hat{\upsilon}_{k,t} + \sigma\phi\left(\hat{\iota}_{k,t}, \delta\right)}{\hat{\mu}_{k,t} - \sigma\phi\left(\hat{\tau}_{k,t}, \delta\right)/\hat{\psi}_{k,t}} \right] \right\}$$

$$\bigcap \left\{ \forall \hat{\tau}_{k,t} \in \mathbb{N}_+, |\hat{\epsilon}_{k,\hat{\tau}_{k,t}}^{IE}| \le \sigma\phi\left(\hat{\tau}_{k,t}, \delta\right) \right\} \bigcap \left\{ \forall \hat{\iota}_{k,t} \in \mathbb{N}_+, |\hat{\epsilon}_{k,\hat{\iota}_{k,t}}^{UE}| \le \sigma\phi\left(\hat{\iota}_{k,t}, \delta\right) \right\}$$

*holds with a probability of at least $1 - \delta$, where:*

$$\hat{\epsilon}_{k,\hat{\tau}_{k,t}}^{IE} = \sum_{i=1}^{t} \epsilon_{k,i} \mathbb{1}\left\{ a_{k,i} \le m_{k,i-1}^l \right\}/\hat{\tau}_{k,t}, \hat{\epsilon}_{k,\hat{\iota}_{k,t}}^{UE} = \sum_{i=1}^{t} \epsilon_{k,i} \mathbb{1}\left\{ a_{k,i} \ge m_{k,i-1}^u \right\}/\hat{\iota}_{k,t}.$$

These lemma implies that our confidence intervals are correct during the learning process for large probability. Let $A = \bigcap_{k=1}^{K} A_k$. A simple union bound inequality shows that $A$ holds with a probability of at least $1 - K\delta$. When the event A happens, all estimators' confidence bounds are correct and the capacity confidence bounds are correct for all $k \in [K]$ and $t \in [T]$, and thus one arm's capacity should be no more than the sum of lower bounds of other arms' capacities. We now can define the capacity confidence lower bound $m_{k,t}^l$ and the upper bound $m_{k,t}^u$ as the end points of the capacity confidence interval of $m_k$, and refined the bounds with the assumption when $A$ happens as:

$$m_{k,t}^l = \max\left\{ \left\lceil \frac{\hat{\upsilon}_{k,t} - \sigma\phi\left(\hat{\iota}_{k,t}, \delta\right)}{\hat{\mu}_{k,t} + \sigma\phi\left(\hat{\tau}_{k,t}, \delta\right)/\hat{\psi}_{k,t}} \right\rceil, 1 \right\}, \tag{10}$$

$$m_{k,t}^u = \min\left\{ \left\lfloor \frac{\hat{\upsilon}_{k,t} + \sigma\phi\left(\hat{\iota}_{k,t}, \delta\right)}{\hat{\mu}_{k,t} - \sigma\phi\left(\hat{\tau}_{k,t}, \delta\right)/\hat{\psi}_{k,t}} \right\rfloor, N - \sum_{i=1,i\neq k}^{K} m_{i,t}^l \right\} \tag{11}$$

Now we compare the arm capacity estimator confidence interval with Wang et al. (2022a):

Wang et al. (2022a): $m_{k,t}^l = \max\left\{ \left\lceil \hat{\upsilon}_{k,t}/\left(\hat{\mu}_{k,t} + \sigma\phi\left(\hat{\tau}_{k,t}, \delta\right) + \sigma\phi\left(\hat{\iota}_{k,t}, \delta\right)\right) \right\rceil, 1 \right\}$

Wang et al. (2022a): $m_{k,t}^u = \min\left\{ \left\lfloor \hat{\upsilon}_{k,t}/\left(\hat{\mu}_{k,t} - \sigma\phi\left(\hat{\tau}_{k,t}, \delta\right) - \sigma\phi\left(\hat{\iota}_{k,t}, \delta\right)\right) \right\rfloor, N - K + 1 \right\}$

Compared with the UCB and LCB in Wang et al. (2022a), one can observe that the key difference between theirs and ours lies in how to handle the estimation error of UE, i.e., the term $\sigma\phi\left(\hat{\iota}_{k,t}, \delta\right)$. Wang et al. (2022a) put it in the denominator, however, we put it above denominator. The reason is that our UCB and LCB is smaller and larger respectively compared to theirs with the same $\hat{\iota}_{k,t}$ and $\hat{\tau}_{k,t}$. So it takes more rounds of UEs and IEs for their confidence intervals to converge. This will be proved by the experiment.

Algorithm 1 states `ActInfCap`, which estimates the arm capacity by adaptively probing the arm with different number of plays for generating samples. More specifically, `ActInfCap` uses the UCB and LCB to generate samples from an arm. The core of `ActInfCap` is the above new confidence interval of arm capacity which is tighter than Wang et al. (2022a). In `ActInfCap`, the UE and IE are conducted in an alternating way and the UCB and LCB of arm capacity approach each other with more utilities returned.

---

**Algorithm 1** ActInfCap($k, T$)

---

1: **Initialize:** $t \leftarrow 0, m_{k,0}^l \leftarrow 1, m_{k,0}^u \leftarrow N$.
2: Do two rounds of initialization, with one UE and one IE respectively.
3: Observe $U_{k,1}$ and $U_{k,2}$. $m_{k,2}^u \leftarrow N, m_{k,2}^l \leftarrow 1, t \leftarrow 2$.
4: **while** $t < T$ and $m_{k,t-1}^l < m_{k,t-1}^u$ **do**
5:     $t \leftarrow t + 1$
6:     **if** $t$ is an odd number **then**
7:         Assign $a_{k,t} \leftarrow m_{k,t-1}^l$ plays to arm $k$
8:         Observe $U_{k,t}$.    Update $m_{k,t}^l, m_{k,t}^u$ via Equation (10) and (11)
9:     **else**
10:         Assign $a_{k,t} \leftarrow m_{k,t-1}^u$ plays to arm $k$
11:         Observe $U_{k,t}$.    Update $m_{k,t}^l, m_{k,t}^u$ via Equation (10) and (11)
12:     **end if**
13: **end while**
14: Return $m_{k,t}^u$

---

**Theorem 2.** *The output of Algorithm 1, i.e., $m_{k,t}^u$ satisfies:*

$$\mathbb{P}\left[\hat{m}_{k,t}^u = m_k \mid t \geq \xi \frac{2\sigma^2}{\mu_k^2} \log\left(\frac{1}{4\delta}\right) + 2\right] \geq 1 - \delta,$$

*where $\xi$ is a universal constant factor independent of model parameters.*

**Remark.** Theorem 2 states that Algorithm 1 has a sample complexity exactly matches the lower bound. This closes the sample complexity gap.

## 5 REGRET LOWER BOUNDS AND SAMPLE EFFICIENT ALGORITHMS

### 5.1 REGRET LOWER BOUNDS

**Theorem 3.** *Given $K$ and $M$, for any learning algorithm or strategy $\pi$, its instance-independent minmax regret lower bound is:*

$$\mathbb{E}\left[Reg\left(T, \pi\right)\right] \geq \frac{\sigma}{64e\sqrt{2}}\sqrt{TK}.$$

**Remark.** Theorem 3 fills in the blank that previous works Wang et al. (2022a) failed to prove instance-independent regret lower bound. It indicates that the minmax regret lower bound has a dependence $\sqrt{K}$ on the number of arms $K$ and a dependence $\sqrt{T}$ on learning horizon $T$. There is no dependence on the arm capacity $m_k$, which aligns with the sample complexity lower bound stated in Theorem (2) and Algorithm 1. Though Theorem 3 is proved by the conventional paradigm Lattimore & Szepesvári (2020), it is technically non-trivial. The key idea is to carefully balance the trade-off between the per-time-slot regret and the difficulty to learn the capacities. If the utility is small, the per-time-slot regret is small. But it is difficult to distinguish the capacities with returned utilities, since the expected returned utilities' gaps are small with the same capacity gaps.

**Theorem 4.** *$K \in \mathbb{N}$, $\{m_k\}_{k\in[K]} \in \mathbb{N}^K$, and $\{\mu_k\}_{k\in[K]} \in \mathbb{R}_+^K$, for any consistent learning strategy $\pi$, it holds*

$$\liminf_{T\to\infty} \frac{\mathbb{E}\left[Reg\left(T, \pi\right)\right]}{\log\left(T\right)} \geq 2\sum_{k=1}^{K} \frac{c\sigma^2}{\mu_k^2}$$

**Remark.** Theorem 4 states that there is a dependence of the instance-dependent regret lower bound on $\mu_k^{-2}$. It implies that the smaller $\mu_k$ is, the harder it is to learn the optimal action. Again, it has no dependence on the arm capacity $m_k$. This does not contradict with Wang et al. (2022a), whose

instance-dependence lower bound's dependence on the arm capacity $m_k$ is $O((\sigma^2 m_k^2 \log T)/\mu_k^2)$. In fact, the above dependence holds under the assumption $\mu_k^2/(\sigma^2 m_k^2) \geq 2$. This condition implies that $(\sigma^2 m_k^2)/\mu_k^2 \leq 1/2$, yielding $(\sigma^2 m_k^2 \log T)/\mu_k^2 \leq 1/2 \log T$. In other words, their instance-dependent regret lower bound has no dependence on $\mu_k$ and $m_k$, and therefore is quite loose. The key idea in the proof is to find a lower bound of the expected number of bad actions during the whole $T$ time slots. .

## 5.2 EFFICIENT EXPLORATION ALGORITHM

**Efficient exploration algorithm.** Algorithm 2 outlines `PC-CapUL`, which is the abbreviation of Prioritized Coordination of Capacities' UCB and LCB. Its key idea is summarized into four folds. (1) **Preventing excessive UEs**(Line 11). At each time slot, we ensure that the historical number of UE is not larger than the number of IE, i.e., $\hat{\tau}_{k,t} \geq \hat{\iota}_{k,t}$. The UE is play-consuming compared with IE, especially at the early time slots when the capacity confidence interval is not leanred well. During the training process, both $\hat{\iota}_{k,t}$ and $\hat{\tau}_{k,t}$ are required to reach their corresponding limits for the algorithm to learn the capacity $m_k$, and these limits is of similar scale as we will show in the proof of the Lemma 5. But if there are not enough plays for all the arms to be played with UE, then some of them are forced to be played with IE, despite the fact that there are already enough IEs on these arms. These compulsory IEs are important source of regret in our problem setting. So it is not wise for us to play an arm with excessive UEs, and the number of IEs is a natural good limit of the number of UEs according to Lemma 5. (2) **Balancing UE and IE**(Line 13). At each time slot $t$, we tend to let as many arms as possible to be played with UEs. The same insight from Lemma 5 reveals that both $\hat{\tau}_{k,t}$ and $\hat{\iota}_{k,t}$ are required to reach their corresponding limits. And it is always easier to do IEs because IEs require fewer plays than UEs. So we should try to focus on meeting the requirement of UEs and make sure that there is at least one UE on certain arms. And this guarantees the ultimate convergence of our algorithm. (3) **Favorable arms win UE first**(Line 14-20). At each time slot $t$, we should let the arms with larger empirical unit utility to have higher priority when deciding the arms to be played with UE if there is not adequate plays for UE on all arms. This design is derived from the insight we discussed in Theorem 4, and this insight is further verified in Lemma 5. The insight is that it is harder to learn the capacity $m_k$ if the unit utility $\mu_k$ is smaller. So we tend to focus on the arms with larger empirical unit utility and play UEs more often on them, in the hope that $\hat{\tau}_{k,t}$ and $\hat{\iota}_{k,t}$ reach their limits within fewer time slots and then there would be no more regret generated on those arms. Another reason is that the larger unit utility of one arm is, the more regret will be generated by IEs on that arm. By rapidly completing learning the capacity of arms with large empirical unit utility, there are less IEs on these arms and consequently less number of potential large amount of regret derived from excessive IEs on these arms. (4) **Stop learning when converges** (Line 12, and Line 24-27). At each time slot $t$, once an arm's capacity upper bound and lower bound meet with each other, there should be no more exploration on that arm. The probability that the estimated capacity is correct can be guaranteed by Lemma 2. And furthermore, we can do explorations more freely on other arms, since there will be no more UE on the arms that we learn well. And this contributes to sooner convergence of all arms' confidence intervals.

**Regret upper bounds.** The following theorems state the regret upper bounds of Algorithm 2.

**Theorem 5.** *The instance-dependent regret upper bound for Algorithm 2 is:*

$$\mathbb{E}\left[REG(T)\right] \leq \sum_{k=1}^{K} \left( \left( \sum_{i=1}^{K} \frac{2304\sigma^2 m_i^2}{\mu_i^2} \log(T) \right) (\mu_k - c) m_k + \frac{1152 m_k^2}{\mu_k^2} \sigma^2 \log(T) cN \right)$$
$$+ \sum_{k=1}^{K} \left( 2K \max(\mu_k m_k, Nc) \right)$$

**Remark.** This upper bound matches the finding we get in the Theorem 4 that an arm's unit utility is an important characteristic modeling the difficulty to learn the arm's capacity. That is, the larger the unit utility is, the more explorations should be done on that arm. The regret upper bound of Wang et al. (2022a) shares the similar terms in our upper bound when bounding the capacities of optimal arms in their setting. This is because we both use UEs and IEs and confidence interval to estimate the arms' capacities. However, in our setting, it is impossible to distinguish the capacities via variance because the perturbations of the returned utility of all arms follow the same distribution. While in

---

**Algorithm 2** `PC-CapUL`

---

1: **Notation:** $\boldsymbol{m}_t^l := (m_{k,t}^l : k \in [K]), \boldsymbol{m}_t^u := (m_{k,t}^u : k \in [K]), \boldsymbol{U}_t := (U_{k,t} : k \in [K])$.
   $\hat{\boldsymbol{\tau}}_t := (\hat{\tau}_{k,t} : k \in [K]), \hat{\boldsymbol{\iota}}_t := (\hat{\iota}_{k,t} : k \in [K]), \hat{\boldsymbol{\mu}}_t := (\hat{\mu}_{k,t} : k \in [K]), \hat{\boldsymbol{v}}_t := (\hat{v}_{k,t} : k \in [K])$.
   $\boldsymbol{Cndt} := (Cndt_k : k \in [K])$ is a binary vector indicating continue exploration (1) or not (0).
   $\boldsymbol{w} := (w_k, k \in [K])$ is a binary vector with entry 1 indicating do IE and 0 indicating do UE.
   $\odot$ denotes the Hadamard product, $\boldsymbol{e}_k$ denotes a unit vector with $k$-th entry being 1.
2: **Initialization:** $\boldsymbol{m}_0^l \leftarrow \mathbf{1}, \boldsymbol{m}_0^u \leftarrow (N - K + 1)\mathbf{1}, \hat{\boldsymbol{\tau}}_0 \leftarrow \mathbf{0}, \hat{\boldsymbol{\iota}}_0 \leftarrow \mathbf{0}, \boldsymbol{Cndt} \leftarrow \mathbf{1}$.
3: **for** $1 \le t \le K$ **do**
4:    The $t$-th arm do UE and all others do IE: $\boldsymbol{w} \leftarrow \mathbf{1} - \boldsymbol{e}_t$
5:    Set the arm assignment as: $\boldsymbol{a}_t \leftarrow (1 - \boldsymbol{w}) \odot \boldsymbol{m}_{t-1}^u + \boldsymbol{w} \odot \boldsymbol{m}_{t-1}^l$.
6:    Observe $\boldsymbol{U}_t$.
7:    Update: $\boldsymbol{m}_t^l \leftarrow \boldsymbol{m}_{t-1}^l, \boldsymbol{m}_t^u \leftarrow \boldsymbol{m}_{t-1}^u, \hat{\boldsymbol{\tau}}_t \leftarrow \hat{\boldsymbol{\tau}}_{t-1} + \boldsymbol{w}, \hat{\boldsymbol{\iota}}_t \leftarrow \hat{\boldsymbol{\iota}}_{t-1} + \mathbf{1} - \boldsymbol{w}, \hat{\boldsymbol{\mu}}_t$ with (6), $\hat{\boldsymbol{v}}_t$ with (7)
8: **end for**
9: **while** $K + 1 \le t \le T$ **do**
10:    **if** $\boldsymbol{Cndt} \ne \mathbf{0}$ **then**
11:       Record the arms whose IE rounds no more than UE rounds: $w_k \leftarrow \mathbb{I}\{\hat{\tau}_{k,t-1} \le \hat{\iota}_{k,t-1}\}, \forall k$.
12:       Record the converged arms: $w_k \leftarrow \mathbb{I}\{Cndt_k = 0\}, \forall k$.
13:       Calculate the capacity needs: $M_{needs} \leftarrow (1 - \boldsymbol{w}) \cdot \boldsymbol{m}_{t-1}^u + \boldsymbol{w} \cdot \boldsymbol{m}_{t-1}^l$.
14:       $\boldsymbol{\ell} \leftarrow$ sort arms based on mean estimation $\hat{\mu}_{k,t-1}$ in descending order with $Cndt_k \ne 0$
15:       **for** $k = 1, \ldots, K$ **do**
16:          **if** $M_{needs} > N$ **then**
17:             The ranked $k$-th arm (with index $\ell_k$) do IE, and update it to the vector $\boldsymbol{w} \leftarrow \boldsymbol{w} + \boldsymbol{e}_{\ell_k}$
18:             Update capacity needs: $M_{needs} \leftarrow (1 - \boldsymbol{w}) \cdot \boldsymbol{m}_{t-1}^u + \boldsymbol{w} \cdot \boldsymbol{m}_{t-1}^l$.
19:          **end if**
20:       **end for**
21:       Set the arm assignment as: $\boldsymbol{a}_t \leftarrow (1 - \boldsymbol{w}) \odot \boldsymbol{m}_{t-1}^u + \boldsymbol{w} \odot \boldsymbol{m}_{t-1}^l$.
22:       Observe $\boldsymbol{U}_t$.
23:       $\hat{\boldsymbol{\tau}}_t \leftarrow \hat{\boldsymbol{\tau}}_{t-1} + \boldsymbol{w}, \hat{\boldsymbol{\iota}}_t \leftarrow \hat{\boldsymbol{\iota}}_{t-1} + \mathbf{1} - \boldsymbol{w}, \hat{\boldsymbol{\mu}}_t$ with (6), $\hat{\boldsymbol{v}}_t$ with (7), $\boldsymbol{m}_t^l$ with (10), $\boldsymbol{m}_t^u$ with (11)
          $Cndt_k \leftarrow \mathbb{I}\{\boldsymbol{m}_{k,t}^l < \boldsymbol{m}_{k,t}^u\}, \forall k$
24:    **else**
25:       Observe $\boldsymbol{U}_t$.
26:       Set the arm assignment as: $\boldsymbol{a}_t \leftarrow \boldsymbol{m}_{t-1}^l, \boldsymbol{m}_t^l \leftarrow \boldsymbol{m}_{t-1}^l, \boldsymbol{m}_t^u \leftarrow \boldsymbol{m}_{t-1}^u$.
27:    **end if**
28: **end while**

---

their setting, the variance of the returned UE utilities on the arm $k$ and arm $i$ is different even if $m_k \mu_k = m_i \mu_i$ as long as $m_k \ne m_i$. With more complicated setting and less usable information in returned utilities, we design the algorithm 2 which shares similar regret upper bounds as those in Wang et al. (2022a), and this implies that their upper bound is loose.

**Theorem 6.** *Upper bound The instance-independent regret upper bound for Algorithm 2 is:*

$$\mathbb{E}\left[REG(T)\right] \le \sigma \sqrt{\left(9216M^3 + 128KM + 1152M^2N\right) M \left(T \log(T)\right)}$$

$$+ \sum_{k=1}^{K} 2K \max\left(\mu_k m_k, Nc\right) + \sum_{k=1}^{K} K \mu_k m_k$$

**Remark.** This upper bound is derived from refining the bound of number of IEs and UEs one arm demanded before it converges. The design of the arms' priority for UEs, which is ranked by empirical unit utility, improves our estimation on the number of IEs a lot. As it is displayed in the figures of the experiments, $K$ and $m_k$ are positive related to the expectation of the regret. There are not significant changes as $N$ varies. And this is not a conflict because we set the movement cost $c$ a small value as $0.1$. Wang et al. (2022a) only proved an instance-dependent regret upper bound.

## 6 EXPERIMENTS

### 6.1 EXPERIMENT SETTING

This section states the experiment setting, including the number of plays, arms, comparison baselines and parameter settings, etc. The capacity of each arm setting: $m_k = 10 + [\ell \times \text{Rand}(0, 1)]$, where $\ell =$

$5, 10, 15, 20$. Number of arms: $K = 10, 20, 30, 40$. Number of plays: $N = M, M + 0.1M, M + 0.2M, M + 0.4M$. Movement cost: $c = 0.2, 0.1, 0.01$, We consider the default parameters unless we mention to vary them explicitly $\ell = 10, K = 20, N = M + 0.1M, c = 0.1$. We conduct simulations to validate the performance of our algorithm and compare it to other algorithms adapted from MAB. We consider three baselines: MP-SE-SA, Orch proposed in Wang et al. (2022a), and a variant of our proposed algorihtm `PC-CapUL-old`, which replaces the our arm capacity estimator with that of Wang et al. (2022a). Other details are shown in the Appendix A.1

## 6.2 IMPACT OF NUMBER OF ARMS

In figure 1a,1b,1c,1d, we set $K$ as $10, 20, 30, 40$ respectively. It is rather obvious that as there is more arms, it takes more exploration for all algorithm to find the true capacities of each arm, as it is indicated in both the lower and upper bound theorems. And for all $K$ values, our algorithms outperform the other two baselines and the one with better estimators converges much quicker than others. In our simulation of 2000 time slots, the regret of Orch in 1a converges to around $4 \times 10^5$ after 700 time slots, which is much slower than ours. There are mainly two reasons for the difference in convergence speed. First, there are much less tries of UEs at the same time slot in Orch for its parsimonious and maladaptive strategy. The UEs are only allowed in even rounds in Orch. In `PC-CapUL-old`, the arm $k$ is played with UE or IE according to how well the $\mu_k$ and $m_k$ are learned. Second, our confidence intervals are more precise, and converge with fewer explorations. Additional experiments are conducted to verify this, with results shown in Appendix A.5

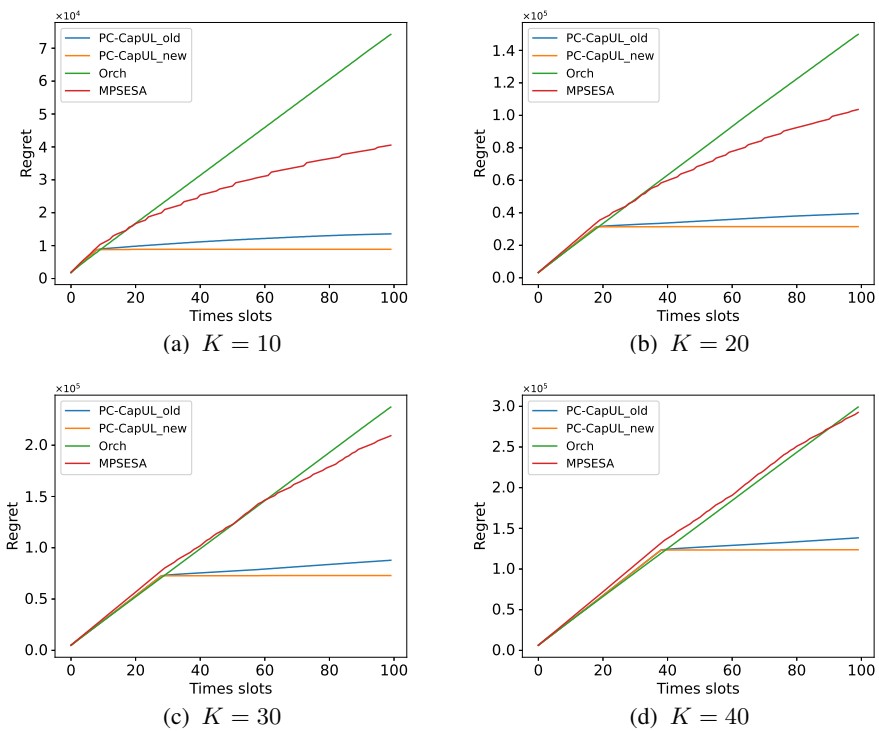

(a) $K = 10$       (b) $K = 20$

(c) $K = 30$       (d) $K = 40$

Figure 1: Impact of number of Arms.

## 7 CONCLUSION

This paper revisits multi-play multi-armed bandit with shareable arm capacities problem. Our result closes the sample complexity gap left by previous works. We also prove new regret lower bounds significantly enhancing previous results. We design an algorithm named `PC-CapUL`, in which we use prioritized coordination of arm capacities upper/lower confidence bound (UCB/LCB) to efficiently balance the exploration vs. exploitation trade-off. We prove both instance-dependent and instance-independent upper bounds for `PC-CapUL`, which match the lower bounds up to some acceptable model-dependent factors. Numerical experiments validate the data efficiency of `PC-CapUL`.

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

# A ADDITIONAL EXPERIMENTS RESULTS

## A.1 ADDITIONAL EXPLANATION ON THE EXPERIMENT SETTINGS

$\mu_k$ is sampled from an even distribution on the interval $[1, 11]$. The utility perturbation $\epsilon$ is set to be of the same Gaussian distribution $\mathcal{N}\left(0, \sigma^2\right)$ for all arms with all settings, and $\sigma = 0.5$. We changed the returned utility function in both Orch and MP-MA-SE algorithm to match our problem setting and compare their performances with ours. We conduct simulations on both versions of our algorithm and the only difference is the estimator of the capacity confidence interval. For every setting we conduct simulations for 20 times and the regrets are averaged.

## A.2 IMPACT OF TOTAL CAPACITY

In figure 2a,2b,2c,2d, we set the interval that $m_k$ is evenly sampled from $[10, 15], [10, 20], [10, 25], [10, 30]$ respectively. We find that as the capacities of arms increase, the regret is larger at the same time slot. There are mainly two reasons:(1) the IEs with only 1 play generates larger regret as the actual capacities increase, and these kind of IE is inevitable in all four algorithms when the capacity confidence intervals are not learned well.(2) It takes more explorations to learn an arm's capacity as the capacity is bigger according to the regret upper bound we get. This result is not contradictory with the finding in the regret lower bound which is unrelated with the capacity, because neither Orch and our algorithm are asserted to be optimal. No matter in what setting , our algorithms outperform the Orch and MP-SE-SA significantly, and the improvement of new estimator is also significant, which leads to much quicker convergence of capacity confidence intervals. In our simulation of 2000 time slots, the regret of Orch in 2a converges to around $1.4 \times 10^6$ after 1750 time slots, which is much slower than ours.

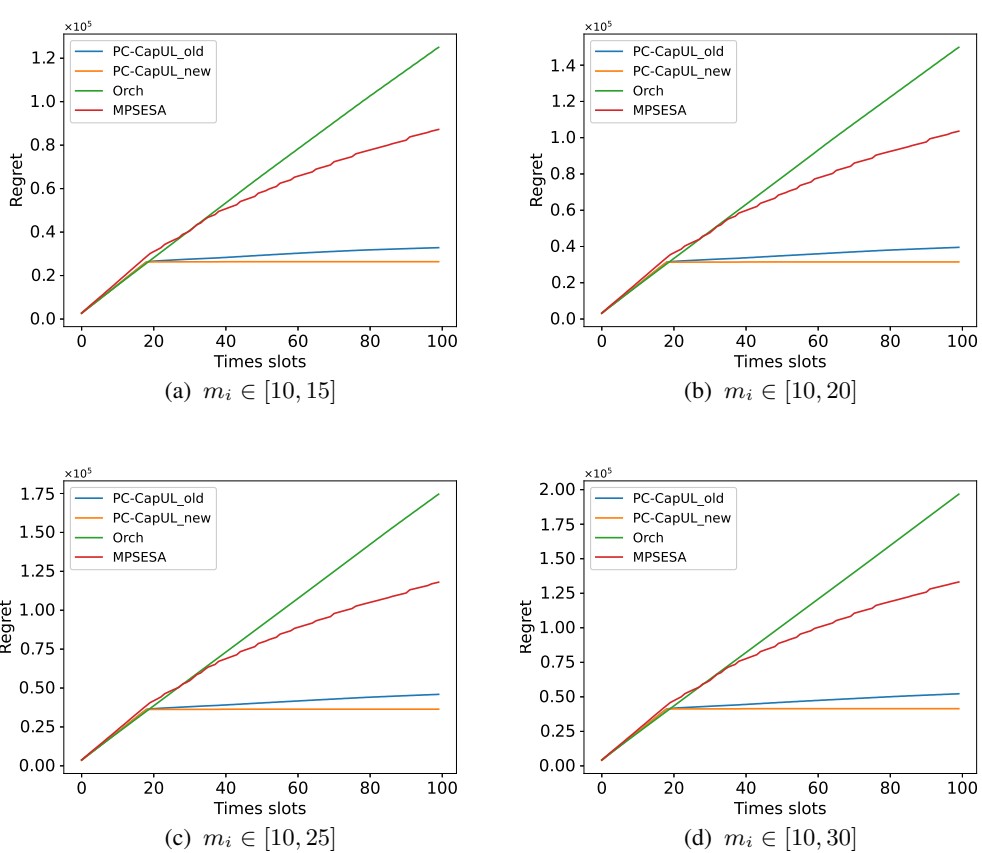

(a) $m_i \in [10, 15]$

(b) $m_i \in [10, 20]$

(c) $m_i \in [10, 25]$

(d) $m_i \in [10, 30]$

Figure 2: Impact of capacities of Arms.

## A.3 IMPACT OF NUMBER OF PLAYS

In figure 3a,3b,3c,3d, we fix $M$ as $\sum_{k=1}^{K} m_k$ and set the ratio $N/M$ as $1, 1.1, 1.2, 1.4$ respectively. We find that as $N$ varies, our algorithms outperform the Orch and the MP-SE-SA in all four settings. The main reason is that the more number of plays, the more UEs we can do at the same time in our algorithms, and consequently the less time slots demanded for the capacity confidence interval to converge. But the increase of plays casts little influence on the performance of Orch, because the UEs in Orch are limited by their conservative strategy, which is designed for the cases when $N < M$.

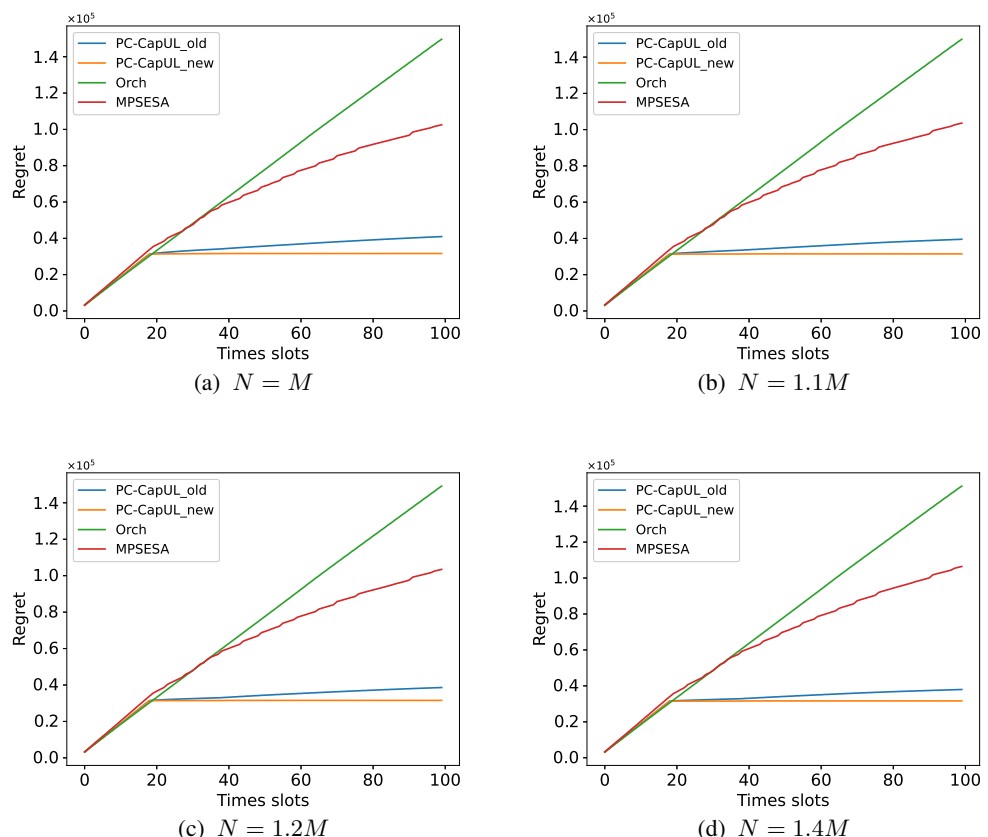

Figure 3: Impact of number of plays

## A.4 IMPACT OF MOVEMENT COST

In figure 4a,4b,4c, we set the movement cost $c = 0.2, 0.1, 0.01$ respectively. We find that as $c$ decreases, the regrets of all four algorithms decrease. It is reasonable that with smaller $c$, the costs of UE become smaller in all four algorithms, and consequently the regret will decrease if other parameters remain unchanged. But this change of movement cost casts little influence on comparison among the regrets of the four algorithms. The main reason is that the movement cost is a significant parameter in the estimation of the regret lower bound but not in the estimation of the the upper bound. The movement cost should be more important and even influence the order of magnitude of the regret if the algorithm has regret upper bound close to the lower bound.

## A.5 COMPARE OF THE OLD AND NEW ESTIMATORS

In figure 5, we set $K = 1$, $M = m_1 = 15$, $N = 30$ , and do UE and IE in an alternating way to explore the capacity. We set the estimators of LCB and UCB of the capacity as (10) and (11) first, and record their values as new-LCB and new-UCB, as shown in the figure 5. And next, we set the

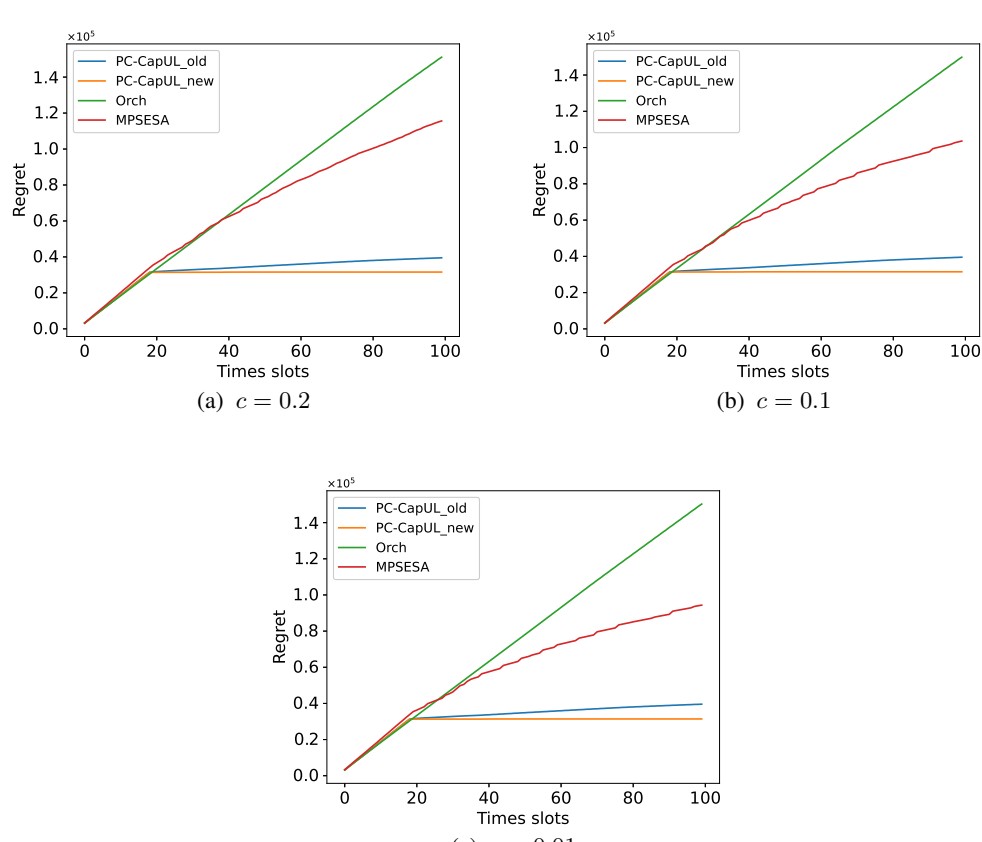

(a) $c = 0.2$        (b) $c = 0.1$

(c) $c = 0.01$

Figure 4: Impact of number of plays

estimators as those used in Wang et al. (2022a), and record their values as old-LCB and old-UCB. In both estimator settings, we conduct simulations for 20 times and the recorded LCB and UCB are averaged. It is quite obvious in the figure 5 that the new estimator converges much more rapidly than the old one, despite the fact that both estimators converge to the correct capacity after adequate explorations.

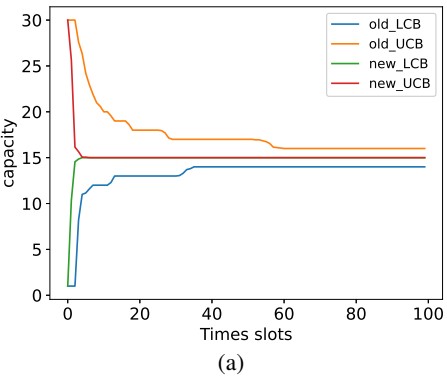

(a)

Figure 5: Impact of number of plays

# B   TECHNICAL PROOFS

## B.1   SAMPLE COMPLEXITY PROOF

**Proof of Theorem 1:** Consider there is an arm with capacity $m_k$ and unit utility value $\mu$. Assume that there are only two possible values for $m_k$: $\{m, m+1\}$ where $m$ is a positive integer, and the perturbation on the arm follows $\mathcal{N}\left(0, \sigma^2\right)$. Let $T$ be the exploration times we do on this arm.

For any strategy $\pi$ that can calculate the capacity after several times of explorations, we consider the probability that the capacity is mistakenly judged,i.e. we consider the probabilities:

$$\mathbb{P}_1\left[\hat{m} = m+1\right]$$
$$\mathbb{P}_2\left[\hat{m} = m\right]$$

where $\hat{m}$ is the estimator given by the strategy $\pi$, and $\mathbb{P}_1$,$\mathbb{P}_2$ are the probability measures defined on the whole $T$ exploration times when the real capacities are $m$ and $m+1$ respectively.

Since there are only two possible values of $m_k$, we have $\{\hat{m} = m+1\} = \{\hat{m} = m\}^C$, meaning that these two events are complementary to each other. This meets the condition of Theorem 14.2 in Lattimore & Szepesvári (2020) and we have:

$$\mathbb{P}_1\left[\hat{m} = m+1\right] + \mathbb{P}_2\left[\hat{m} = m\right]$$
$$\geq \frac{1}{2}\exp\left(-KL\left(\mathbb{P}_1, \mathbb{P}_2\right)\right)$$

As for the KL-divergence, we use the result we get in (17). Let $N\left(T\right)$ be the number of actions assigned by $\pi$ satisfying that $a_t \geq m+1$, and then we have:

$$KL\left(\mathbb{P}_1, \mathbb{P}_2\right) = \mathbb{E}_1\left[N\left(T\right)\right]\frac{\mu^2}{2\sigma^2} \leq T\frac{\mu^2}{2\sigma^2}$$

If $\pi$ works well for probability at least $\delta$, then we have:

$$\mathbb{P}_1\left[\hat{m} = m+1\right] + \mathbb{P}_2\left[\hat{m} = m\right] \leq 2\delta$$

And consequently we get:

$$2\delta$$
$$\geq \mathbb{P}_1\left[\hat{m} = m+1\right] + \mathbb{P}_2\left[\hat{m} = m\right]$$
$$\geq \frac{1}{2}\exp\left(-KL\left(\mathbb{P}_1, \mathbb{P}_2\right)\right)$$
$$\geq \frac{1}{2}\exp\left(-T\frac{\mu^2}{2\sigma^2}\right)$$

By rearranging the terms we get:

$$T \geq \frac{2\sigma^2}{\mu^2}\log\left(\frac{1}{4\delta}\right)$$

∎

**Proof of Theorem 2:** We first assume that the capacity falls into the confidence set, to ensure that the counters $\hat{\tau}_{k,t}$ and $\hat{\iota}_{k,t}$ are correct. This lead to the confidence set for the reward mean:

$$\mathbb{P}[\forall t, \mu_k - \sigma\phi\left(\hat{\tau}_{k,t}, \delta\right)/\hat{\psi}_{k,t} \leq \hat{\mu}_{k,t} \leq \mu_k + \sigma\phi\left(\hat{\tau}_{k,t}, \delta\right)/\hat{\psi}_{k,t}] \geq 1-\delta$$
$$\mathbb{P}[\forall t, m_k\mu_k - \sigma\phi\left(\hat{\iota}_{k,t}, \delta\right) \leq \hat{v}_{k,t} \leq m_k\mu_k + \sigma\phi\left(\hat{\iota}_{k,t}, \delta\right)] \geq 1-\delta$$

If the reward means satisfy

$$\mu_k - \sigma\phi\left(\hat{\tau}_{k,t}, \delta\right)/\hat{\psi}_{k,t} \leq \hat{\mu}_{k,t} \leq \mu_k + \sigma\phi\left(\hat{\tau}_{k,t}, \delta\right)/\hat{\psi}_{k,t}$$
$$m_k\mu_k - \sigma\phi\left(\hat{\iota}_{k,t}, \delta\right) \leq \hat{v}_{k,t} \leq m_k\mu_k + \sigma\phi\left(\hat{\iota}_{k,t}, \delta\right)$$

It leads to that
$$m_k \in [m_{k,t}^l, m_{k,t}^u].$$
The chicken-egg problem with reward means and capacities is resolved by the fact that
$$m_k \in [1, N].$$
Thus, we use $1, N$ to initialize $m_{k,t}^l, m_{k,t}^u$ respectively
$$m_{k,0}^l = 1, m_{k,0}^u = N$$
This initialization makes the $\hat{v}_{k,1}$ and $\hat{\mu}_{k,1}$ fall into the above inequalities with the reward gathered by the initialized correct lower and upper bound of capacity. And the valid $\hat{v}_{k,1}$ and $\hat{\mu}_{k,1}$ leads to the subsequent valid updates of $m_{k,1}^l$ and $m_{k,1}^u$, which enable us to collect new valid observations in the next round. Doing this recursively, we resolve the chicken-egg problem. We next focus on the case that all the reward mean and capacity inequalities hold and ignore the small probability of $2\delta$ that at least one of them fails.

We first derive a lower bound on $m_{k,t}^l$ as

$$
\begin{aligned}
m_{k,t}^l &= \max\left\{\left\lceil \frac{\hat{v}_{k,t} - \sigma\phi\left(\hat{\iota}_{k,t}, \delta\right)}{\hat{\mu}_{k,t} + \sigma\phi\left(\hat{\tau}_{k,t}, \delta\right)/\hat{\psi}_{k,t}} \right\rceil, 1\right\} \\
&\geq \frac{\hat{v}_{k,t} - \sigma\phi\left(\hat{\iota}_{k,t}, \delta\right)}{\hat{\mu}_{k,t} + \sigma\phi\left(\hat{\tau}_{k,t}, \delta\right)/\hat{\psi}_{k,t}} \\
&\geq \frac{m_k\mu_k - 2\sigma\phi\left(\hat{\iota}_{k,t}, \delta\right)}{\mu_k + 2\sigma\phi\left(\hat{\tau}_{k,t}, \delta\right)/\hat{\psi}_{k,t}} \\
&= m_k - 2\frac{m_k\sigma\phi\left(\hat{\tau}_{k,t}, \delta\right)/\hat{\psi}_{k,t} + \sigma\phi\left(\hat{\iota}_{k,t}, \delta\right)}{\mu_k + 2\sigma\phi\left(\hat{\tau}_{k,t}, \delta\right)/\hat{\psi}_{k,t}} \\
&\geq m_k - 2\frac{m_k\sigma\phi\left(\hat{\tau}_{k,t}, \delta\right)/\hat{\psi}_{k,t} + \sigma\phi\left(\hat{\iota}_{k,t}, \delta\right)}{\mu_k}
\end{aligned}
$$

We next derive an upper bound on $m_{k,t}^u$ as:

$$
\begin{aligned}
m_{k,t}^u &= \min\left\{\left\lfloor \frac{\hat{v}_{k,t} + \sigma\phi\left(\hat{\iota}_{k,t}, \delta\right)}{\hat{\mu}_{k,t} - \sigma\phi\left(\hat{\tau}_{k,t}, \delta\right)/\hat{\psi}_{k,t}} \right\rfloor, N\right\} \\
&\leq \frac{\hat{v}_{k,t} + \sigma\phi\left(\hat{\iota}_{k,t}, \delta\right)}{\hat{\mu}_{k,t} - \sigma\phi\left(\hat{\tau}_{k,t}, \delta\right)/\hat{\psi}_{k,t}} \\
&\leq \frac{m_k\mu_k + 2\sigma\phi\left(\hat{\iota}_{k,t}, \delta\right)}{\mu_k - 2\sigma\phi\left(\hat{\tau}_{k,t}, \delta\right)/\hat{\psi}_{k,t}} \\
&\leq m_k + 2\frac{m_k\sigma\phi\left(\hat{\tau}_{k,t}, \delta\right)/\hat{\psi}_{k,t} + \sigma\phi\left(\hat{\iota}_{k,t}, \delta\right)}{\mu_k - 2\sigma\phi\left(\hat{\tau}_{k,t}, \delta\right)/\hat{\psi}_{k,t}}
\end{aligned}
$$

The above inequality holds when $\mu_k - 2\sigma\phi\left(\hat{\tau}_{k,t}, \delta\right)/\hat{\psi}_{k,t} > 0$. A sufficient condition is:
$$\phi\left(\hat{\tau}_{k,t}, \delta\right) < 0.25\mu_k/\sigma. \tag{12}$$
We will discuss how to guarantee (12) later. Suppose that (12) holds, then it follows that

$$
\begin{aligned}
&m_{k,t}^u - m_{k,t}^l \\
&= 2\frac{m_k\sigma\phi\left(\hat{\tau}_{k,t}, \delta\right)/\hat{\psi}_{k,t} + \sigma\phi\left(\hat{\iota}_{k,t}, \delta\right)}{\mu_k - 2\sigma\phi\left(\hat{\tau}_{k,t}, \delta\right)/\hat{\psi}_{k,t}} + 2\frac{m_k\sigma\phi\left(\hat{\tau}_{k,t}, \delta\right)/\hat{\psi}_{k,t} + \sigma\phi\left(\hat{\iota}_{k,t}, \delta\right)}{\mu_k} \\
&\leq 4\frac{m_k\sigma\phi\left(\hat{\tau}_{k,t}, \delta\right)/\hat{\psi}_{k,t} + \sigma\phi\left(\hat{\iota}_{k,t}, \delta\right)}{\mu_k} + 2\frac{m_k\sigma\phi\left(\hat{\tau}_{k,t}, \delta\right)/\hat{\psi}_{k,t} + \sigma\phi\left(\hat{\iota}_{k,t}, \delta\right)}{\mu_k} \\
&= 6\frac{m_k\sigma\phi\left(\hat{\tau}_{k,t}, \delta\right)/\hat{\psi}_{k,t} + \sigma\phi\left(\hat{\iota}_{k,t}, \delta\right)}{\mu_k}
\end{aligned}
$$

To reveal the true arm capacity, a sufficient condition is:

$$6\frac{m_k\sigma\phi\left(\hat{\tau}_{k,t},\delta\right)/\hat{\psi}_{k,t}+\sigma\phi\left(\hat{\iota}_{k,t},\delta\right)}{\mu_k}<1 \tag{13}$$

Under our alternating of UE and IE algorithm, we have that when $t$ is an even number, $\hat{\tau}_{k,t}=\hat{\iota}_{k,t}$. This implies that

$$\phi\left(\hat{\tau}_{k,t},\delta\right)=\phi\left(\hat{\iota}_{k,t},\delta\right).$$

Then, (13) is equivalent to

$$\phi\left(\hat{\iota}_{k,t},\delta\right)<\frac{1}{6}\frac{\mu_k}{\sigma}\frac{\hat{\psi}_{k,t}}{m_k+\hat{\psi}_{k,t}}. \tag{14}$$

We next prove that $\hat{\psi}_{k,t}$ has nice lower bound under certain conditions. Given an arbitrary constant $\gamma\in(0,1)$, a sufficient condition to guarantee $m_{k,t}^l>\gamma m_k$ is:

$$2\frac{m_k\sigma\phi\left(\hat{\tau}_{k,t},\delta\right)/\hat{\psi}_{k,t}+\sigma\phi\left(\hat{\iota}_{k,t},\delta\right)}{\mu_k}<(1-\gamma)m_k$$

When $t$ is an even number, this is equivalent to

$$\phi\left(\hat{\iota}_{k,t},\delta\right)<\frac{1-\gamma}{6}\frac{\mu_k}{\sigma}\frac{\hat{\psi}_{k,t}m_k}{m_k+\hat{\psi}_{k,t}}\Leftarrow\phi\left(\hat{\iota}_{k,t},\delta\right)<\frac{1-\gamma}{6}\frac{\mu_k}{\sigma}\frac{m_k}{m_k+1}.$$

A refined sufficient condition is:

$$\phi\left(\hat{\iota}_{k,t},\delta\right)<\frac{1-\gamma}{12}\frac{\mu_k}{\sigma}. \tag{15}$$

Let $t_\gamma$ denote the minimum $t$ satisfying (15):

$$t_\gamma:=\arg\min_{t>0}\phi\left(t,\delta\right)<\frac{1-\gamma}{12}\frac{\mu_k}{\sigma}.$$

Consider a positive number $\beta>0$, it holds that

$$t>2(\beta+1)t_\gamma\Rightarrow\hat{\psi}_{k,t}\geq\frac{t_\gamma+\gamma m_k\beta t_\gamma}{(\beta+1)t_\gamma}=\frac{1+\gamma\beta m_k}{\beta+1}\geq\frac{\gamma\beta}{\beta+1}m_k.$$

If the true capacity is identified before $2(\beta+1)t_\gamma$ rounds, then we have that the sample complexity is $2(\beta+1)t_\gamma$. If not, then applying (14) the lower bound of $\hat{\psi}_{k,t}$ implies a refined sufficient condition to identify the true capacity

$$\phi\left(\hat{\iota}_{k,t},\delta\right)<\frac{1}{6}\frac{\mu_k}{\sigma}\frac{\frac{\gamma\beta}{\beta+1}m_k}{m_k+\frac{\gamma\beta}{\beta+1}m_k}\Leftrightarrow\phi\left(\hat{\iota}_{k,t},\delta\right)<\frac{1}{6}\frac{\mu_k}{\sigma}\frac{\gamma\beta}{\beta+1+\gamma\beta}. \tag{16}$$

Thus the sample complexity is

$$\arg\min_{t>0}\phi\left(\hat{\iota}_{k,t},\delta\right)<\frac{\mu_k}{\sigma}\xi$$

where $\xi$ is a constant defined as

$$\xi:=\min_{\beta>0,\gamma\in(0,1)}\max\left\{\frac{1}{6}\frac{\gamma\beta}{\beta+1+\gamma\beta},\frac{(\beta+1)(1-\gamma)}{6},0.25\right\}$$

Noticing that in the first two rounds of explorations, we assign 1 and $N$ plays to the arm respectively, so a constant 2 should be added on the upper bound. This proof is then complete. ∎

## B.2 REGRET LOWER BOUND PROOF

**Proof of Theorem 3:** To avoid unnecessary mathematical subtleties and simplify the proof, we focus on the case that $M/K$ is an integer and $K/4$ is also an integer. We first contract two instances of the problem as follows:

- Instance $E_1$: each arm whose index is an odd number has $\left(\frac{M}{K} - 1\right)$ units of capacity and each of the remaining arms has $\left(\frac{M}{K} + 1\right)$ units of capacity. The per unit reward mean is fixed to $\mu$, i.e., $\mu_1 = \ldots = \mu_K = \mu$, and variance is fixed to $\sigma$, i.e., $\sigma_1 = \ldots = \sigma_K = \sigma$. Formally,

$$
\begin{array}{ccccc}
 & \text{arm 1} & \text{arm 2} & \text{arm } K-1 & \text{arm } K \\
\text{Instance } E_1: & M/K - 1 & M/K + 1 & \cdots \quad M/K - 1 & M/K + 1 \\
 & \mu, \sigma & \mu, \sigma & \mu, \sigma & \mu, \sigma
\end{array}
$$

- Instance $E_2$: each arm whose index is an even number has $\left(\frac{M}{K} - 1\right)$ units of capacity and each of the remaining arms has $\left(\frac{M}{K} + 1\right)$ units of capacity. The per unit reward mean is fixed to $\mu$, i.e., $\mu_1 = \ldots = \mu_K = \mu$, and variance is fixed to $\sigma$, i.e., $\sigma_1 = \ldots = \sigma_K = \sigma$. Formally,

$$
\begin{array}{ccccc}
 & \text{arm 1} & \text{arm 2} & \text{arm } K-1 & \text{arm } K \\
\text{Instance } E_2: & M/K + 1 & M/K - 1 & \cdots \quad M/K + 1 & M/K - 1 \\
 & \mu, \sigma & \mu, \sigma & \mu, \sigma & \mu, \sigma
\end{array}
$$

For an arbitrary learning algorithm or strategy $\pi$, let $R_T(\pi, E_1)$ and $R_T(\pi, E_2)$ denote $\pi$'s regrets in instance $E_1$ and $E_2$ respective. Let $T_1$ denote the number of time slots that at least $\frac{K}{4}$ arms with odd index are assigned exactly $\left(\frac{M}{K} - 1\right)$ plays. Let $A$ denote the event that $T_1 \geq \frac{1}{2}T$:

$$
A = \left\{ T_1 \geq \frac{1}{2}T \right\}.
$$

We can use event $A$ to bound the expectation of the regret in $E_1$ as follows:

$$
\begin{aligned}
& \mathbb{E}\left[R_T(\pi, E_1)\right] \\
= & \mathbb{E}\left[R_T(\pi, E_1)\,\mathbb{1}\{A\}\right] + \mathbb{E}\left[R_T(\pi, E_1)\,\mathbb{1}\{A^C\}\right] \\
\geq & 0 + \frac{TK}{8}\min(\mu - c, c)\,\mathbb{P}_{E_1}\left(A^C\right).
\end{aligned}
$$

And similarly we have

$$
\mathbb{E}\left[R_T(\pi, E_2)\right] \geq \frac{TK}{8} \cdot 2(\mu - c)\,\mathbb{P}_{E_2}(A).
$$

Note that the Theorem 14.2 in Lattimore & Szepesvári (2020) indicates:

$$
\mathbb{P}_{E_1}\left(A^C\right) + \mathbb{P}_{E_2}(A) \geq \frac{1}{2}\exp\left(-KL\left(\mathbb{P}_{E_1}, \mathbb{P}_{E_2}\right)\right).
$$

Then, the sum of the regrets of $\pi$ in two instances can be lower bounded as:

$$
\begin{aligned}
& \mathbb{E}\left[R_T(\pi, E_1)\right] + \mathbb{E}\left[R_T(\pi, E_2)\right] \\
\geq & \frac{TK}{8}\min(\mu - c, c)\left(\mathbb{P}_{E_1}\left(A^C\right) + \mathbb{P}_{E_2}(A)\right) \\
\geq & \frac{TK}{16}\min(\mu - c, c)\exp\left(-KL\left(\mathbb{P}_{E_1}, \mathbb{P}_{E_2}\right)\right).
\end{aligned}
$$

Note that the probability measure $\mathbb{P}_{E_1}$ is defined on the entire learning process of $T$ time slots, i.e.

$$
\mathbb{P}_{E_1}\left[\boldsymbol{a}_1, \boldsymbol{x}_1, ..., \boldsymbol{a}_T, \boldsymbol{x}_T\right] = \prod_{t=1}^{T} \pi_t\left(\boldsymbol{a}_t | \boldsymbol{a}_1, \boldsymbol{x}_1, ..., \boldsymbol{a}_{T-1}, \boldsymbol{x}_{T-1}\right) P_{E_1, \boldsymbol{a}_t}\left(\boldsymbol{x}_t\right),
$$

where $\boldsymbol{a}_t$ is the action chosen at the time slot $t$ and vector $\boldsymbol{x}_t$ is the resulting reward on the $K$ arms after playing $\boldsymbol{a}_t$. $\pi_t$ is the probability measure of the action $\boldsymbol{a}_t$ after the observation of the past $t - 1$

sets of actions and rewards, and $P_{E_1, \boldsymbol{a}_t}$ is the probability measure of the reward vector $\boldsymbol{x}_t$ for fixed action $\boldsymbol{a}_t$ in instance $E_1$. As for the calculation of the KL-divergence, we can separate it into $T$ actions.

$$KL\left(\mathbb{P}_{E_1}, \mathbb{P}_{E_2}\right)$$

$$= \mathbb{E}_{E_1}\left[\log\left(\frac{d\mathbb{P}_{E_1}}{d\mathbb{P}_{E_2}}\right)\right]$$

$$= \mathbb{E}_{E_1}\left[\sum_{t=1}^{T} \log \frac{P_{E_1, \boldsymbol{a}_t}\left(\boldsymbol{x}_t\right)}{P_{E_2, \boldsymbol{a}_t}\left(\boldsymbol{x}_t\right)}\right]$$

$$= \sum_{t=1}^{T} \mathbb{E}_{E_1}\left[\log \frac{P_{E_1, \boldsymbol{a}_t}\left(\boldsymbol{x}_t\right)}{P_{E_2, \boldsymbol{a}_t}\left(\boldsymbol{x}_t\right)}\right]$$

$$= \sum_{t=1}^{T} \mathbb{E}_{E_1}\left[\mathbb{E}_{E_1}\left[\log \frac{P_{E_1, \boldsymbol{a}_t}\left(\boldsymbol{x}_t\right)}{P_{E_2, \boldsymbol{a}_t}\left(\boldsymbol{x}_t\right)}\middle| \boldsymbol{a}_t\right]\right]$$

$$= \sum_{t=1}^{T} \mathbb{E}_{E_1}\left[KL\left(P_{E_1, \boldsymbol{a}_t}, P_{E_2, \boldsymbol{a}_t}\right)\right]$$

where in the last equality we use that under $\mathbb{P}_{E_1}\left(\cdot|\boldsymbol{a}_t\right)$ the distribution of $\boldsymbol{x}_t$ is $P_{E_1, \boldsymbol{a}_t}$.

Because the measure $P_{E_1, \boldsymbol{a}_t}$ is a product of $K$ independent probability measures, we can decompose the KL divergence as follows:

$$KL\left(P_{E_1, \boldsymbol{a}_t}, P_{E_2, \boldsymbol{a}_t}\right) = \sum_{k=1}^{K} KL\left(P_{E_1, a_{k,t}}, P_{E_2, a_{k,t}}\right)$$

where $P_{E_1, a_{k,t}}$ and $P_{E_2, a_{k,t}}$ follow normal distribution:

$$P_{E_1, a_{k,t}} \sim \mathcal{N}\left(\quad \min\left(a_{k,t}, m_k^{(1)}\right)\mu - a_{k,t} \cdot c \quad, \quad \sigma^2 \quad\right)$$

$$P_{E_2, a_{k,t}} \sim \mathcal{N}\left(\quad \min\left(a_{k,t}, m_k^{(2)}\right)\mu - a_{k,t} \cdot c \quad, \quad \sigma^2 \quad\right),$$

and $m_k^{(1)}$ and $m_k^{(2)}$ denote the capacities of arm $k$ in the $E_1$ and $E_2$ respectively. There is a formula about the KL-divergence of two Gaussian distributions:

**Lemma 3.** *For each $i \in \{1, 2\}$, let $\mu_i \in \mathbb{R}, \sigma_i^2 > 0$ and $P_i = \mathcal{N}\left(\mu_i, \sigma_i^2\right)$. Then we have:*

$$KL\left(P_1, P_2\right) = \frac{1}{2}\left(\log\left(\frac{\sigma_2^2}{\sigma_1^2}\right) + \frac{\sigma_1^2}{\sigma_2^2} - 1\right) + \frac{\left(\mu_1 - \mu_2\right)^2}{2\sigma_2^2}$$

Applying lemma 3, we have:

$$KL\left(P_{E_1, a_{1,t}}, P_{E_2, a_{1,t}}\right) = \frac{\left(\min\left(a_{1,t}, m_k^{(1)}\right)\mu - \min\left(a_{1,t}, m_k^{(2)}\right)\mu\right)^2}{2\sigma^2}$$

We want to find the action $a_{1,t}$ maximizing $KL\left(P_{E_1, a_{1,t}}, P_{E_2, a_{1,t}}\right)$ at time slot $t$ on the first arm. It is easy to find that $a_{1,t}$ should be no less than $m_1^{(2)} = \frac{M}{K} + 1$ so that $KL\left(P_{E_1, a_{1,t}}, P_{E_2, a_{1,t}}\right)$ reaches its maximal. The same is true for other arms $k$ with odd $k$. And similarly we should let the action $a_{2,t} \geq m_2^{(1)} = \frac{M}{K} + 1$ in order to let $KL\left(P_{E_1, a_{2,t}}, P_{E_2, a_{2,t}}\right)$ reaches its maximal. The same is true for other arms $k$ with even $k$. So we get that:

$$KL\left(P_{E_1, a_{1,t}}, P_{E_2, a_{1,t}}\right) \leq \frac{2\mu^2}{\sigma^2}$$

$$KL\left(P_{E_1, a_{2,t}}, P_{E_2, a_{2,t}}\right) \leq \frac{2\mu^2}{\sigma^2}$$

It should be noted that it is possible $a_{1,t}, a_{2,t}, ..., a_{K,t}$ can not be taken at the same time in the real world. But there is no conflict since we are only interested in the upper bound of the KL-divergence.

Note that $\mathbb{E}[X] \leq \max[X]$, then we get:

$$KL\left(\mathbb{P}_{E_1}, \mathbb{P}_{E_2}\right)$$

$$=\sum_{t=1}^{T} \mathbb{E}_{E_1}\left[KL\left(P_{E_1,\boldsymbol{a}_t}, P_{E_2,\boldsymbol{a}_t}\right)\right]$$

$$\leq T \cdot \max_{\boldsymbol{a}\in\mathcal{A}}\left[KL\left(P_{E_1,\boldsymbol{a}}, P_{E_2,\boldsymbol{a}}\right)\right]$$

$$=T \cdot \max_{\boldsymbol{a}\in\mathcal{A}}\left[\sum_{k=1}^{K} KL\left(P_{E_1,a_k}, P_{E_2,a_k}\right)\right]$$

$$\leq T \cdot \sum_{k=1}^{K} \max_{a_k\in[N]}\left[KL\left(P_{E_1,a_k}, P_{E_2,a_k}\right)\right]$$

$$\leq T \cdot \sum_{k=1}^{K} \frac{2\mu^2}{\sigma^2}$$

$$=TK\frac{2\mu^2}{\sigma^2}$$

Furthermore, by letting $c = \frac{1}{2}\mu$, we have that:

$$\mathbb{E}\left[R_T\left(\pi, E_1\right)\right] + \mathbb{E}\left[R_T\left(\pi, E_2\right)\right]$$

$$\geq \frac{TK}{16} \min\left(\mu - c, c\right)\exp\left(-KL\left(\mathbb{P}_{E_1}, \mathbb{P}_{E_2}\right)\right)$$

$$=\frac{TK}{32}\mu\exp\left(-KL\left(\mathbb{P}_{E_1}, \mathbb{P}_{E_2}\right)\right)$$

$$\geq \frac{TK}{32}\mu\exp\left(-2TK\frac{\mu^2}{\sigma^2}\right)$$

We let $\mu = \sigma/\sqrt{2TK}$ and then we get

$$\max\left(\quad \mathbb{E}\left[R_T\left(\pi, E_1\right)\right], \quad \mathbb{E}\left[R_T\left(\pi, E_2\right)\right]\quad\right) \geq \frac{\sigma}{32e\sqrt{2}}\sqrt{TK}$$

This proof is then complete. ∎

**Proof of Theorem 4:** Here we only consider the set of algorithms that is consistent over the class of MP-MAB $\mathcal{E}$ we described in section 2, and we further require that the perturbation of the returned utility follows the Gaussian distribution $\mathcal{N}\left(0, \sigma^2\right)$ for simplicity, where $\sigma^2 \leq 1/2$ .

A policy $\pi$ is defined as consistent over a class of bandits $\mathcal{E}'$ if for all $E \in \mathcal{E}'$ and $p > 0$ that :

$$\lim_{T\to\infty} \frac{REG\left(T\right)}{T^p} = 0$$

First we choose a consistent policy $\pi$. Let $E_1 \in \mathcal{E}$ be an instance, and there are $m_k$ units of capacities with unit utility $\mu_k$ on the arm $k$. Next we will consider the number of time slots $TB_k\left(T\right)$ when the arm $k$ is assigned with more than $m_k$ plays by $\pi$ in $T$ time slots, i.e.

$$TB_k\left(T\right) := \sum_{t=1}^{T} \mathbb{1}\left\{a_{k,t} \geq m_k + 1\right\}$$

For fixed $k \in [K]$, let $E_2 \in \mathcal{E}$ be another instance, and for $j \neq k$, there are $m_j$ units of capacities with unit utility $\mu_j$ on the arm $j$. On the arm $k$ in $E_2$, there are $m_k + 1$ units of capacities with unit

utility $\mu_j$. Let $A$ be the event that $TB_k \leq \frac{T}{2}$:

$$A := \left\{ TB_k \leq \frac{T}{2} \right\}$$

Let $R_T(\pi, E_1), R_T(\pi, E_2)$ denote the policy $\pi$'s regret in instance $E_1$ and $E_2$. Then by similar analysis in previous subsection, we have:

$$\mathbb{E}\left[R_T(\pi, E_1)\right]$$
$$= \mathbb{E}\left[R_T(\pi, E_1) \mathbb{1}\{A\}\right] + \mathbb{E}\left[R_T(\pi, E_1) \mathbb{1}\{A^C\}\right]$$
$$\geq 0 + \frac{T}{2} c \mathbb{P}_{E_1}\left(A^C\right)$$

Then similarly we have :

$$\mathbb{E}\left[R_T(\pi, E_2)\right] \geq \frac{T}{2}(\mu_k - c)\, \mathbb{P}_{E_2}(A)$$

Then the sum of the regrets of $\pi$ in two instances can be lower bounded as:

$$\mathbb{E}\left[R_T(\pi, E_1)\right] + \mathbb{E}\left[R_T(\pi, E_2)\right]$$
$$\geq \frac{T}{2} \min(\mu_k - c, c)\left(\mathbb{P}\left(A^C\right) + \mathbb{P}(A)\right)$$
$$\geq \frac{T}{4} \min(\mu_k - c, c) \exp\left(-KL\left(\mathbb{P}_{E_1}, \mathbb{P}_{E_2}\right)\right)$$

As for the KL-divergence, we can decompose it by time slots and arms as it is shown in the previous subsection:

$$KL\left(\mathbb{P}_{E_1}, \mathbb{P}_{E_2}\right)$$
$$= \sum_{t=1}^{T} \mathbb{E}_{E_1}\left[KL\left(P_{E_1,\boldsymbol{a}_t}, P_{E_2,\boldsymbol{a}_t}\right)\right]$$
$$= \sum_{t=1}^{T} \mathbb{E}_{E_1}\left[\sum_{i=1}^{K} KL\left(P_{E_1,a_{i,t}}, P_{E_2,a_{i,t}}\right)\right]$$

And note that $E_1$ and $E_2$ are the same only except the arm $k$. Thus the above equality can be reduced to:

$$\sum_{t=1}^{T} \mathbb{E}_{E_1}\left[\sum_{i=1}^{K} KL\left(P_{E_1,a_{i,t}}, P_{E_2,a_{i,t}}\right)\right]$$
$$= \sum_{t=1}^{T} \mathbb{E}_{E_1}\left[KL\left(P_{E_1,a_{k,t}}, P_{E_2,a_{k,t}}\right)\right]$$
$$= \sum_{t=1}^{T} \mathbb{E}_{E_1}\left[KL\left(P_{E_1,a_{k,t}}, P_{E_2,a_{k,t}}\right) \mathbb{1}\{a_{k,t} \geq m_k + 1\}\right]$$
$$+ \sum_{t=1}^{T} \mathbb{E}_{E_1}\left[KL\left(P_{E_1,a_{k,t}}, P_{E_2,a_{k,t}}\right) \mathbb{1}\{a_{k,t} \leq m_k\}\right]$$
$$= \sum_{t=1}^{T} \mathbb{E}_{E_1}\left[KL\left(P_{E_1,a_{k,t}}, P_{E_2,a_{k,t}}\right) \mathbb{1}\{a_{k,t} \geq m_k + 1\}\right] + 0$$

According to lemma 3, when $a_{k,t} \geq m_k + 1$:

$$KL\left(P_{E_1,a_{k,t}}, P_{E_2,a_{k,t}}\right) = \frac{\mu_k^2}{2\sigma^2}$$

Thus we have :

$$\sum_{t=1}^{T} \mathbb{E}_{E_1} \left[ KL \left( P_{E_1, a_{k,t}}, P_{E_2, a_{k,t}} \right) \mathbb{1} \left\{ a_{k,t} \geq m_k + 1 \right\} \right]$$

$$= \sum_{t=1}^{T} \mathbb{E}_{E_1} \left[ \mathbb{1} \left\{ a_{k,t} \geq m_k + 1 \right\} \right] \frac{\mu_k^2}{2\sigma^2}$$

$$= \mathbb{E}_{E_1} \left[ \sum_{t=1}^{T} \mathbb{1} \left\{ a_{k,t} \geq m_k + 1 \right\} \right] \frac{\mu_k^2}{2\sigma^2}$$

$$= \mathbb{E}_{E_1} \left[ TB_k \left( T \right) \right] \frac{\mu_k^2}{2\sigma^2}$$

Consequently we calculate the KL-divergence as :

$$KL \left( \mathbb{P}_{E_1}, \mathbb{P}_{E_2} \right) = \mathbb{E}_{E_1} \left[ TB_k \left( T \right) \right] \frac{\mu_k^2}{2\sigma^2} \tag{17}$$

Then we have:

$$\mathbb{E} \left[ R_T \left( \pi, E_1 \right) \right] + \mathbb{E} \left[ R_T \left( \pi, E_2 \right) \right] \geq \frac{T}{4} \min \left( \mu_k - c, c \right) \exp \left( -\mathbb{E}_{E_1} \left[ TB_k \left( T \right) \right] \frac{\mu_k^2}{2\sigma^2} \right)$$

Rearranging and taking the limit inferior on $T$ leads to:

$$\liminf_{T \to \infty} \frac{\mathbb{E}_{E_1} \left[ TB_k \left( T \right) \right]}{\log \left( T \right)} \geq \frac{2\sigma^2}{\mu_k^2} \liminf_{T \to \infty} \frac{\log \left( \frac{T \min(\mu_k - c, c)}{4(\mathbb{E}[R_T(\pi, E_1)] + \mathbb{E}[R_T(\pi, E_2)])} \right)}{\log \left( T \right)}$$

$$= \frac{2\sigma^2}{\mu_k^2} \left( 1 - \limsup_{T \to \infty} \frac{\log \left( \mathbb{E} \left[ R_T \left( \pi, E_1 \right) \right] + \mathbb{E} \left[ R_T \left( \pi, E_2 \right) \right] \right)}{\log \left( T \right)} \right)$$

Since the policy $\pi$ is consistent, then for any $p > 0$ there is a constant $C_p$ that for sufficiently large $T$: $\mathbb{E} \left[ R_T \left( \pi, E_1 \right) \right] + \mathbb{E} \left[ R_T \left( \pi, E_2 \right) \right] \leq C_p T^p$, which implies that:

$$\limsup_{T \to \infty} \frac{\log \left( \mathbb{E} \left[ R_T \left( \pi, E_1 \right) \right] + \mathbb{E} \left[ R_T \left( \pi, E_2 \right) \right] \right)}{\log \left( T \right)}$$

$$\leq \limsup_{T \to \infty} \frac{p \log \left( T \right) + \log \left( C_p \right)}{\log \left( T \right)}$$

$$= p$$

Since $p$ can be arbitrarily small, we have

$$\limsup_{T \to \infty} \frac{\log \left( \mathbb{E} \left[ R_T \left( \pi, E_1 \right) \right] + \mathbb{E} \left[ R_T \left( \pi, E_2 \right) \right] \right)}{\log \left( T \right)} = 0$$

And consequently,

$$\liminf_{T \to \infty} \frac{\mathbb{E}_{E_1} \left[ TB_k \left( T \right) \right]}{\log \left( T \right)} \geq \frac{2\sigma^2}{\mu_k^2}$$

It should be noted that

$$\mathbb{E}\left[R_T\left(\pi, E_1\right)\right]$$

$$=\mathbb{E}_{E_1}\left[\sum_{t=1}^{T}\left(f\left(\boldsymbol{a}^*\right)-f\left(\boldsymbol{a}_t\right)\right)\right]$$

$$=\mathbb{E}_{E_1}\left[\sum_{t=1}^{T}\sum_{k=1}^{K}\left[\left(m_k\mu_k - cm_k\right)-\left(\min\left\{a_{k,t}, m_k\right\}\cdot\mu_k - c\cdot a_{k,t}\right)\right]\right]$$

$$=\mathbb{E}_{E_1}\left[\sum_{k=1}^{K}\sum_{t=1}^{T}\left[\left(m_k\mu_k - cm_k\right)-\left(\min\left\{a_{k,t}, m_k\right\}\cdot\mu_k - c\cdot a_{k,t}\right)\right]\right]$$

$$\geq\mathbb{E}_{E_1}\left[\sum_{k=1}^{K}\sum_{t=1}^{T}\left[\left(m_k\mu_k - cm_k\right)-\left(\min\left\{a_{k,t}, m_k\right\}\cdot\mu_k - c\cdot a_{k,t}\right)\right]\mathbb{1}\left\{a_{k,t}\geq m_k+1\right\}\right]$$

$$\geq\mathbb{E}_{E_1}\left[\sum_{k=1}^{K}\sum_{t=1}^{T}c\cdot\mathbb{1}\left\{a_{k,t}\geq m_k+1\right\}\right]$$

$$=c\cdot\sum_{k=1}^{K}\mathbb{E}_{E_1}\left[TB_k\left(T\right)\right]$$

Taking the limit inferior on $T$ leads to:

$$\liminf_{T\to\infty}\frac{\mathbb{E}\left[R_T\left(\pi, E_1\right)\right]}{\log\left(T\right)}$$

$$\geq c\cdot\sum_{k=1}^{K}\liminf_{T\to\infty}\frac{\mathbb{E}_{E_1}\left[TB_k\left(T\right)\right]}{\log\left(T\right)}$$

$$\geq c\cdot\sum_{k=1}^{K}\frac{2\sigma^2}{\mu_k^2}$$

And the proof is complete. ∎

### B.3 REGRET UPPER BOUND PROOF

Before proving Theorem 5, we need to prove two Lemmas first.

**Proof of Lemma 1**

Consider the confidence interval for $\mu_k$. Because

$$\hat{\mu}_{k,t} - \mu_k$$

$$=\frac{\sum_{s=1}^{t}\left(U_{k,s}\left(a_{k,s}\right)+c\cdot a_{k,s}\right)\mathbb{1}\left\{a_{k,s}\leq m_{k,s-1}^{l}\right\}}{\sum_{s=1}^{t}a_{k,s}\mathbb{1}\left\{a_{k,s}\leq m_{k,s-1}^{l}\right\}}-\mu_k$$

$$=\frac{\sum_{s=1}^{t}\left(\min\left\{a_{k,s}, m_k\right\}\cdot\mu_k - c\cdot a_{k,s}+\epsilon_{k,s}+c\cdot a_{k,s}\right)\mathbb{1}\left\{a_{k,s}\leq m_{k,s-1}^{l}\right\}}{\sum_{s=1}^{t}a_{k,s}\mathbb{1}\left\{a_{k,s}\leq m_{k,s-1}^{l}\right\}}-\mu_k$$

When the event $A_k$ defined in Lemma 2 happens, then for time slot $s$ satisfying $a_{k,s}\leq m_{k,s-1}^{l}$, we have that the action $a_{k,s}\leq m_k$.

And thus we get

$$\hat{\mu}_{k,t} - \mu_k$$

$$= \frac{\sum_{s=1}^{t} \left( \min\{a_{k,s}, m_k\} \cdot \mu_k - c \cdot a_{k,s} + \epsilon_{k,s} + c \cdot a_{k,s} \right) \mathbb{1}\left\{ a_{k,s} \leq m_{k,s-1}^l \right\}}{\sum_{s=1}^{t} a_{k,s} \mathbb{1}\left\{ a_{k,s} \leq m_{k,s-1}^l \right\}} - \mu_k$$

$$= \frac{\sum_{s=1}^{t} \left( a_{k,s} \cdot \mu_k + \epsilon_{k,s} \right) \mathbb{1}\left\{ a_{k,s} \leq m_{k,s-1}^l \right\}}{\sum_{s=1}^{t} a_{k,s} \mathbb{1}\left\{ a_{k,s} \leq m_{k,s-1}^l \right\}} - \mu_k$$

$$= \frac{\sum_{s=1}^{t} \epsilon_{k,s} \mathbb{1}\left\{ a_{k,s} \leq m_{k,s-1}^l \right\}}{\sum_{s=1}^{t} a_{k,s} \mathbb{1}\left\{ a_{k,s} \leq m_{k,s-1}^l \right\}}$$

$$= \frac{\hat{\tau}_{k,t}}{\sum_{s=1}^{t} a_{k,s} \mathbb{1}\left\{ a_{k,s} \leq m_{k,s-1}^l \right\}} \cdot \frac{\sum_{s=1}^{t} \epsilon_{k,s} \mathbb{1}\left\{ a_{k,s} \leq m_{k,s-1}^l \right\}}{\hat{\tau}_{k,t}}$$

$$= \frac{\hat{\tau}_{k,t}}{\sum_{s=1}^{t} a_{k,s} \mathbb{1}\left\{ a_{k,s} \leq m_{k,s-1}^l \right\}} \cdot \hat{\epsilon}_{k,\hat{\tau}_{k,t}}^{IE}$$

By rearranging the the equality above, we get the following statement if $A_k$ happens:

$$\frac{\sum_{s=1}^{t} a_{k,s} \mathbb{1}\left\{ a_{k,s} \leq m_{k,s-1}^l \right\}}{\hat{\tau}_{k,t}} \left( \hat{\mu}_{k,t} - \mu_k \right) \in \left[ -\sigma\phi\left(\hat{\tau}_{k,t}, \delta\right), \sigma\phi\left(\hat{\tau}_{k,t}, \delta\right) \right]$$

Note that $\hat{\psi}_{k,t}$ is defined as:

$$\hat{\psi}_{k,t} = \frac{\sum_{s=1}^{t} a_{k,s} \mathbb{1}\left\{ a_{k,s} \leq m_{k,s-1}^l \right\}}{\hat{\tau}_{k,t}}$$

We get that

$$\left( \hat{\mu}_{k,t} - \mu_k \right) \in \left[ -\sigma\phi\left(\hat{\tau}_{k,t}, \delta\right) / \hat{\psi}_{k,t}, \sigma\phi\left(\hat{\tau}_{k,t}, \delta\right) / \hat{\psi}_{k,t} \right]$$

and consequently we get the confidence interval for $\mu_k$ as:

$$\mu_k \in \left[ \hat{\mu}_{k,T^*} - \sigma\phi\left(\hat{\tau}_{k,T^*}, \delta\right) / \hat{\psi}_{k,t}, \hat{\mu}_{k,T^*} + \sigma\phi\left(\hat{\tau}_{k,T^*}, \delta\right) / \hat{\psi}_{k,t} \right]$$

Next we consider the confidence interval of $m_k \mu_k$ when $A_k$ happens:

$$\hat{\upsilon}_{k,T^*} - m_k \mu_k$$

$$= \frac{\sum_{s=1}^{T^*} \left( \min\{a_{k,s}, m_k\} \cdot \mu_k - c \cdot a_{k,s} + \epsilon_{k,s} + c \cdot a_{k,s} \right) \mathbb{1}\left\{ a_{k,s} \geq m_{k,s-1}^u \right\}}{\hat{\iota}_{k,T^*}} - m_k \mu_k$$

$$= \frac{\sum_{s=1}^{T^*} \left( m_k \mu_k + \epsilon_{k,s} \right) \mathbb{1}\left\{ a_{k,s} \geq m_{k,s-1}^u \right\}}{\hat{\iota}_{k,T^*}} - m_k \mu_k$$

$$= \frac{\sum_{s=1}^{T^*} \epsilon_{k,s} \mathbb{1}\left\{ a_{k,s} \geq m_{k,s-1}^u \right\}}{\hat{\iota}_{k,T^*}}$$

$$= \hat{\epsilon}_{k,\hat{\iota}_{k,T^*}}^{UE}$$

And similarly we get the confidence interval of $m_k\mu_k$:

$$m_k\mu_k \in [\hat{v}_{k,T^*} - \sigma\phi\left(\hat{\imath}_{k,T^*}, \delta\right), \hat{v}_{k,T^*} + \sigma\phi\left(\hat{\imath}_{k,T^*}, \delta\right)]$$

Thus we know that for fixed $k$, for all $t$, these confidence intervals are correct with probability $\mathbb{P}\{A_k\}$, and in the proof of Lemma 2, we will show that $\mathbb{P}\{A_k\} \geq 1 - \delta$.

**Proof of Lemma 2**

We first display the concentration inequality we use:

**Lemma 4.** *(Bourel et al. (2020),Lemma 5) Let $Y_i, ..., Y_t$ be a sequence of $t$ i.d.d real-valued random variables with mean $\mu$, such that $Y_t - \mu$ is $\sigma$-sub-Gaussian. Let $\mu_t = \frac{1}{t}\sum_{s=1}^{t} Y_s$ be the empirical mean estimate. Then, for all $\sigma \in (0, 1)$, it holds*

$$\mathbb{P}\left(\exists t \in \mathbb{N}, |\mu_t - \mu| \geq \sigma\sqrt{\left(1 + \frac{1}{t}\right)\frac{2\log\left(\sqrt{t + 1}/\delta\right)}{t}}\right) \leq \delta$$

The key challenge is to handle the chicken-egg problem that the confidence interval of the arm capacity relies on the estimation of the utility mean and the estimation of the utility mean relies on the estimation of the arm capacity to distinguish UEs and IEs. Misleading UEs as IEs would make the reward mean estimation incorrect.

To understand the chicken-egg problem, let us consider a simple problem sharing the essence of our problem:

$$X_i = q_i\mu + \epsilon_i,$$

where $\epsilon_i$'s are independent $\sigma$-sub-Gaussian random variable. Let $q_i'$ denote our guess of $q_i$, which may or may not equal to $q_i$. We use $q_i'$ to estimate $\mu$. The estimator aligned with us is:

$$\hat{\mu}_t = \frac{\sum_i^t X_i}{\sum_i^t q_i'}.$$

Then it follows that

$$\begin{aligned}
\hat{\mu}_t - \mu &= \frac{\sum_i^t q_i\mu + \epsilon_i}{\sum_i^t q_i'} - \mu \\
&= \frac{\sum_i^t q_i\mu + \epsilon_i - \mu\sum_i^t q_i'}{\sum_i^t q_i'} \\
&= \frac{\sum_i^t q_i\mu - \mu\sum_i^t q_i'}{\sum_i^t q_i'} + \frac{\sum_i^t \epsilon_i}{\sum_i^t q_i'} \\
&= \frac{\sum_i^t q_i\mu - \mu\sum_i^t q_i'}{\sum_i^t q_i'} + \frac{t}{\sum_i^t q_i'}\frac{\sum_i^t \epsilon_i}{t}.
\end{aligned}$$

Then it follows that

$$\left|\hat{\mu}_t - \mu - \mathrm{Err}_t\right| = \left|\frac{t}{\sum_i^t q_i'}\frac{\sum_i^t \epsilon_i}{t}\right| = \frac{t}{\sum_i^t q_i'}\left|\frac{\sum_i^t \epsilon_i}{t}\right|,$$

where

$$\mathrm{Err}_t := \frac{\sum_i^t q_i\mu - \mu\sum_i^t q_i'}{\sum_i^t q_i'}$$

denotes the mis-classification error. Then letting $Y_i \leftarrow \epsilon_i$, $t \leftarrow \hat{\tau}_{k,t}$ and $\delta \leftarrow \delta/2$ in Lemma 4, and applying Lemma 4, we have that

$$\mathbb{P}\left[\forall t, \left|\frac{\sum_i^{\hat{\tau}_{k,t}} \epsilon_i}{\hat{\tau}_{k,t}}\right| \leq \sigma\phi(\hat{\tau}_{k,t}, \delta)\right] \geq 1 - \delta/2.$$

This implies the following confidence interval:

$$\mathbb{P}[\forall t, |\hat{\mu}_t - \mu - \text{Err}_t| \le \sigma \phi(\hat{\tau}_{k,t}, \delta)] \ge 1 - \delta/2.$$

This implies that under mis-classification of $q_i$ a uniform confidence interval still holds, but one needs to adjust the bound of the interval with the mis-specification error $\text{Err}_t$.

With the above argument in mind, we know that if there are mistakes in the confidence bounds of capacity, the following uniform confidence interval should hold by adjusting the bound with mis-classification error.

$$\mathbb{P}[\forall t, \mu_k - \sigma \phi(\hat{\tau}_{k,t}, \delta)/\hat{\psi}_{k,t} - \text{Err}_t \le \hat{\mu}_{k,t} \le \mu_k + \sigma \phi(\hat{\tau}_{k,t}, \delta)/\hat{\psi}_{k,t} + \text{Err}_t] \ge 1 - \delta/2,$$

Let us now go back to the chicken problem. With the analysis above, let us consider the good event falls into to the $1 - \delta/2$ probability region, such that

$$\mu_k - \sigma \phi(\hat{\tau}_{k,t}, \delta)/\hat{\psi}_{k,t} - \text{Err}_t \le \hat{\mu}_{k,t} \le \mu_k + \sigma \phi(\hat{\tau}_{k,t}, \delta)/\hat{\psi}_{k,t} + \text{Err}_t$$

holds for all $t$. We next solve the chiken-egg problem by showing that $\text{Err}_t = 0$. Note that $m_k \in [1, N - K + 1]$ is known as a prior. In the initialization rounds, the UE is conducted by $N - K + 1$ and IE is conducted by 1, namely.

$$m_{k,0}^l = 1, m_{k,0}^u = N - K + 1.$$

This initialization generates no initialization error. Thus, with the reward obtained from the initialization to update the confidence, we would have $\text{Err}_t = 0$. This zero error, would lead to the updated estimation of the confidence interval of the arm capacity being correct, as it is implied from the confidence of the utility mean estimation. Thus with the updated confidence interval, we would do correct UE and IE. Doing this recursively, we would have $\text{Err}_t = 0$.

And with similar analysis we know that there is also no mis-classifications of UEs if the sampled perturbations $\epsilon_{k,t}$ on the UE utilities satisfy the condition we desctibed in Lemma 2 that for $\forall \hat{\iota}_{k,t} \in \mathbb{N}_+., |\hat{\epsilon}_{k,\hat{\iota}_{k,t}}^{UE}| \le \sigma \phi(\hat{\iota}_{k,t}, \delta)$. And we know that according to Lemma 4, this condition holds with probability more than $1 - \delta/2$ as well. Thus by Union-Bound inequality we know that $\mathbb{P}\{A_k\} \ge 1 - \delta$. Then the Lemma 2 and Lemma 1 are proved ∎

**Proof of Theorem 5**.

Before proving the upper bound of the regret, we first find the maximal number of UEs and IEs for an arm's capacity interval to converge in another form.

**Lemma 5.** *For any arm $k$, time slot $t$, and $0 < \delta \le \min\left(2exp\left(-1152 m_k^2 \sigma^2/\mu_k^2\right), 2\sqrt{T+1}\right)$, if the number of IEs $\hat{\tau}_{k,t}$ and UEs $\hat{\iota}_{k,t}$ are both no less than $\frac{1152 m_k^2 \sigma^2 \log(2/\delta)}{\mu_k^2}$, then*

$$\mathbb{P}\left(m_{k,t}^l = m_{k,t}^u | \hat{\tau}_{k,t}, \hat{\iota}_{k,t} \ge \frac{1152 m_k^2 \sigma^2 \log(2/\delta)}{\mu_k^2}\right) \ge 1 - \delta$$

Since (13) is a sufficient condition for the confidence interval to converge when $\phi(\hat{\tau}_{k,t}, \delta) < 0.25 \mu_k/\delta$, and notice that $\hat{\psi}_{k,t} \ge 1$, then we have that:

$$6 \frac{m_k \sigma \phi(\hat{\tau}_{k,t}, \delta) + \sigma \phi(\hat{\iota}_{k,t}, \delta)}{\mu_k} < 1$$

is also a sufficient condition. And a simple case to meet this condition is that:

$$\phi(\hat{\tau}_{k,t}, \delta) \le \frac{\mu_k}{12 \sigma m_k} \quad , \quad \phi(\hat{\iota}_{k,t}, \delta) \le \frac{\mu_k}{12 \sigma}$$

And this case also meets the requirement that $\phi(\hat{\tau}_{k,t}, \delta) < 0.25 \mu_k/\delta$ because $m_k \ge 1$. Solving the inequalities above, we get that:

$$\hat{\tau}_{k,t} \ge \frac{1152 \sigma^2 m_k^2 \log(2/\delta)}{\mu_k^2} \quad , \quad \hat{\iota}_{k,t} \ge \frac{1152 \sigma^2 \log(2/\delta)}{\mu_k^2}$$

is a sufficient condition for the capacity confidence interval to converge with the assumptions that $\sqrt{\hat{\tau}_{k,t}+1} \leq 2/\delta$ and $\sqrt{\hat{\iota}_{k,t}+1} \leq 2/\delta$. This assumption is right naturally since we will set $\delta = 2/T$ eventually.

It should be noted that $\phi(t,\delta)$ is monotonically decreasing for $t > 0$, and thus excessive explorations will not make a converged capacity confidence interval contain more than two integers at future time slots.

When most of the arms' capacities are learnt, the rest of the arms can freely be played with UEs or IEs because there are probably enough plays. Since in `PC-CapUL` 2 it is only required that $\hat{\iota}_{k,t} \leq \hat{\tau}_{k,t}$, there may be excessive UEs because the the requirement of number of UEs is $m_k$ times smaller than the number of IEs for arm $k$.

So after $\frac{1152\sigma^2 m_k^2 \log(2/\delta)}{\mu_k^2}$ UEs and IEs, we have $m_{k,t}^l = m_{k,t}^u$. And the lemma 5 is proved.

When the event $A$ happens, the capacity confidence intervals on all arms at all time slots $t > K$ are correct. Here we define an IE or UE at at time slot $t$ as an "effective" one when

$$\hat{\tau}_{k,t} \leq \frac{1152m_k^2\sigma^2 \log(2/\delta)}{\mu_k^2} \quad \text{or} \quad \hat{\iota}_{k,t} \leq \frac{1152m_k^2\sigma^2 \log(2/\delta)}{\mu_k^2},$$

and as a "wasted" IE or UE when

$$\hat{\tau}_{k,t} > \frac{1152m_k^2\sigma^2 \log(2/\delta)}{\mu_k^2} \quad \text{or} \quad \hat{\iota}_{k,t} > \frac{1152m_k^2\sigma^2 \log(2/\delta)}{\mu_k^2},$$

And there is no wasted UEs in our algorithm: since $\hat{\iota}_{k,t} \leq \hat{\tau}_{k,t}$, if there is a wasted UE, there should also be a wasted IE, and then the requirement of lemma 5 is met, which means there should be no increase in $\hat{\iota}_{k,t}$ and leads to a contradiction. Let

$$G(\delta) := \sum_{k=1}^{K} \frac{1152m_k^2\sigma^2 \log(2/\delta)}{\mu_k^2}$$

be the number of most time slots we need to meet the requirement of $\hat{\iota}_{k,t}$ for all $k$ according to lemma 5. Assume that there is no effective IEs in these $G(\delta)$ time slots, and thus we need at most another $G(\delta)$ time slots to do effective IEs. So after $2G(\delta)$ time slots, we have both

$$\hat{\iota}_{k,t}, \hat{\tau}_{k,t} \geq \frac{1152m_k^2\sigma^2 \log(2/\delta)}{\mu_k^2},$$

which meets the requirement of lemma 5. And there will be no more UE or IE attempt after $2G(\delta)$ time slots because all the confidence intervals converge to integer values.

For an arm k, there is at most $2G(\delta)$ time slots for IE and at most $\frac{1152m_k^2\sigma^2 \log(2/\delta)}{\mu_k^2}$ time slots for UE.

We now know the maximal numbers of both IE and UE for the capacity confidence interval to converge to an integer for each arm. Next we will see how the numbers of IE and UE affect the regret $REG(T)$.

We can recalculate $REG\,(T)$ arm by arm:

$$REG\,(T)$$

$$= \sum_{t=1}^{T} \left(f\left(\mathbf{a}^*\right) - f\left(\mathbf{a}_t\right)\right)$$

$$= \sum_{t=1}^{T} \left(\left(\sum_{k=1}^{K} \left(m_k \mu_k - c m_k\right)\right) - \left(\sum_{k=1}^{K} \left(\min\{a_{k,t}, m_k\} \cdot \mu_k - c \cdot a_{k,t}\right)\right)\right)$$

$$= \sum_{t=1}^{T} \left(\sum_{k=1}^{K} \left(m_k \mu_k - c m_k - \min\left\{a_{k,t}, m_k\right\} \cdot \mu_k + c \cdot a_{k,t}\right)\right)$$

$$= \sum_{k=1}^{K} \left(\sum_{t=1}^{T} \left(m_k \mu_k - c m_k - \min\left\{a_{k,t}, m_k\right\} \cdot \mu_k + c \cdot a_{k,t}\right)\right)$$

$$= \sum_{k=1}^{K} REG_k\,(T)$$

where $REG_k\,(T) := \sum_{t=1}^{T} \left(m_k \mu_k - c m_k - \min\left\{a_{k,t}, m_k\right\} \cdot \mu_k + c \cdot a_{k,t}\right)$

And then the expectation of $REG_k\,(T)$ can be divided by the event $A$:

$$\mathbb{E}\left[REG_k\,(T)\right]$$

$$= \mathbb{E}\left[REG_k\,(T)\,\mathbb{1}\{A\}\right] + \mathbb{E}\left[REG_k\,(T)\,\mathbb{1}\{A^C\}\right]$$

$$\leq \mathbb{E}\left[REG_k\,(T)\,\mathbb{1}\{A\}\right] + \mathbb{P}\left(A^C\right)\max\left(\mathbb{E}\left[REG_k\,(T)\right]\right)$$

The second term can be bounded by $T$ multiply the maximum of the per-time-slot regret on the arm $k$, which can be generated by either IE with only one play or UE with all $N$ plays. So let $Regmax_k$ be the maximal per-time-slot regret we get on arm $k$, so we have $Regmax_k \leq \max\left(m_k \mu_k, Nc\right)$ is a constant value. And thus the second term can be bounded by $(K\delta)\,T \cdot Regmax_k$.

As for the first term, we know that as $A$ happens, the algorithm works well and the capacity confidence interval converges to the true capacity $m_k$ after $2G\,(\delta)$ time slots, and there will be no regret for the following time slots. Thus we can bound the first term if the numbers of UE and IE on arm k is bounded. For the UE on arm k, the regret is at most $(N - m_k)\,c$ when all the plays are assigned to arm k, and for the IE, the regret is at most $(m_k - 1)\,(\mu_k - c)$ when there is only one play assigned to arm k. Then we can relate the first term with the expectation of numbers of IE and UE as:

$$\mathbb{E}\left[REG_k\,(T)\,\mathbb{1}\{A\}\right]$$

$$\leq \mathbb{E}\left[\hat{\tau}_{k,T}\right](m_k - 1)\,(\mu_k - c) + \mathbb{E}\left[\hat{\iota}_{k,T}\right](N - m_k)\,c$$

$$\leq \mathbb{E}\left[\hat{\tau}_{k,T}\right]m_k\,(\mu_k - c) + \mathbb{E}\left[\hat{\iota}_{k,T}\right]Nc$$

Then consequently we can bound the expectation of the regret with the following lemma:

**Lemma 6.** *In our problem setting, the expectation of regret is related with the expectation of numbers of IE and UE on each arm as:*

$$\mathbb{E}\left[REG\,(T)\right]$$

$$= \sum_{k=1}^{K} \mathbb{E}\left[REG_k\,(T)\right]$$

$$\leq \sum_{k=1}^{K} \left(\mathbb{E}\left[REG_k\,(T)\,\mathbb{1}\{A\}\right] + \mathbb{P}\left(A^C\right)\max\left(\mathbb{E}\left[REG_k\,(T)\right]\right)\right)$$

$$\leq \sum_{k=1}^{K} \left(\mathbb{E}\left[\hat{\tau}_{k,T}\right]m_k\,(\mu_k - c) + \mathbb{E}\left[\hat{\iota}_{k,T}\right]Nc + \mathbb{P}\left(A^C\right)\max\left(\mathbb{E}\left[REG_k\,(T)\right]\right)\right)$$

$$\leq \sum_{k=1}^{K} \left(\mathbb{E}\left[\hat{\tau}_{k,T}\right]m_k\,(\mu_k - c) + \mathbb{E}\left[\hat{\iota}_{k,T}\right]Nc + KT\delta Regmax_k\right)$$

We first consider a rough bound derived from the above inequality, where we set the expectation of both $\hat{\tau}_{k,T}$ and $\hat{\iota}_{k,T}$ to the maximum as $2G(\delta)$ and $\frac{1152m_k^2\sigma^2\log(2/\delta)}{\mu_k^2}$. A refined bound is also proposed as Theorem 7. By letting $\delta = \frac{2}{T}$, $M$ be the number of plays and $c$ be the movement cost, the sum of the regret is bound by:

$$\mathbb{E}[REG(T)] \leq \sum_{k=1}^{K}\left(\left(\sum_{i=1}^{K}\frac{2304\sigma^2m_i^2}{\mu_i^2}\log(T)\right)(\mu_k - c)m_k + \frac{1152m_k^2}{\mu_k^2}\sigma^2\log(T)cN\right)$$

$$+ \sum_{k=1}^{K}\left(\frac{2}{T}KT\cdot Regmax_k\right)$$

$$\leq \left(\sum_{k=1}^{K}\mu_k m_k\right)\left(\sum_{i=1}^{K}\frac{2304m_i^2}{\mu_i^2}\right)\sigma^2\log(T) + \sum_{k=1}^{K}\left(\frac{1152m_k^2}{\mu_k^2}\sigma^2\log(T)cN\right)$$

$$+ \sum_{k=1}^{K}2K\cdot Regmax_k$$

Then the Theorem 5 is proved. ∎

**Proof of Theorem 6**.

As it is shown in the regret expectation upper bound above, for the arm k, if the average reward $\mu_k$ is significantly small, then the regret can be outrageously large. The main reason is that the $\mathbb{E}[\hat{\tau}_{k,T}]$ of the arms with large average reward should be much smaller than $2G(\delta)$ according to PC-CapUL 2, since the capacity confidence intervals on these arms should converge more rapidly than others, and then there should be no more UEs or IEs on these arms in subsequent time slots. In PC-CapUL 2 the empirical unit reward $\hat{\mu}_{k,t}$ serves as an estimator predicting how much regret we will get at one single time slot, and we decide the action $\mathbf{a}_t$ according to the rank of $\{\hat{\mu}_{k,t}\}_{k\in[K]}$. However, the choice of the estimator is not unique, and one can use $\hat{\mu}_{k,t}m_{k,t}^u$ or other estimators as well. In this algorithm and the proof of its regret upper bound, it is shown that $\hat{\mu}_{k,t}$ is a qualified estimator. Following the idea we mention above, we will refine the bound of $\mathbb{E}[\hat{\tau}_{k,T}]$ with the following lemma:

**Lemma 7.** *Fixed arm $k$, and for another arm $i$ with $\mu_i < \mu_k$. consider the number of time slots in the training process of* PC-CapUL *2 when the arm $i$ is played with UE but the arm $k$ is played with IE and the IE on arm $k$ is not compulsory because of the lack of IEs. We let $Ac_{k,i}$ be the number of such time slots, and then we have :*

$$Ac_{k,i} \leq \frac{32\sigma^2\log(T)}{(\mu_k - \mu_i)^2} + 1$$

We first prove the Lemma 7.

Let $T^*$ be the last time slot that the arm $i$ is played with UE but the arm $k$ is played with IE and the IE on arm $k$ is not compulsory because of the lack of IEs. Then we know that from the $K+1$ time slot to the $T^* - 1$ time slot, there is at least $Ac_{k,i} - 2$ time slots at which the arm $i$ is played with UE and arm $k$ is played with IE. Since we know that the arm $i$ is played with UE at time slot $T^*$, and in PC-CapUL 2 the arm $i$ cannot be played with more UEs than IEs, then there must be at least $Ac_{k,i} - 2$ time slots at which the arm $i$ is played with IEs. Summing up these $Ac_{k,i} - 2$ time slots with the at least 1 time slots in initialization phase when the arm $i$ is forced to be played by IEs. We know that before $T^*$, the arm $i$ is played with at least $Ac_{k,i} - 1$ IEs. And the same is true for arm $k$.

Then at time slot $T^*$, since the arm $k$ is not forced to be played with IE, then we must have that the arm $i$ is chosen to be played with UE for its higher empirical unit utility $\hat{\mu}_{i,T^*}$. Consequently we

have $\hat{\mu}_{i,T^*} \geq \hat{\mu}_{k,T^*}$, which is only possible when the lower bound of $\hat{\mu}_{k,T^*}$ is not larger than the upper bound of $\hat{\mu}_{i,T^*}$. Then we have:

$$\mu_k - \sigma\phi\left(Ac_{k,i} - 1, \frac{2}{T}\right)\bigg/\hat{\psi}_{k,t} \leq \mu_i + \sigma\phi\left(Ac_{k,i} - 1, \frac{2}{T}\right)\bigg/\hat{\psi}_{k,t}$$

Notice the fact that $\hat{\psi}_{k,t} \geq 1$. By solving the above inequality we get the lemma:

$$Ac_{k,i} \leq \frac{32\sigma^2 \log(T)}{(\mu_k - \mu_i)^2} + 1$$

The lemma is then proved.

For the arm k, we now divide the IE into 3 groups:(1) the IEs caused by the UEs of other arms with unit utility no less than $\frac{1}{2}\mu_k$.(2) the IEs caused by the UE of other arms with unit utility less than $\frac{1}{2}\mu_k$.(3) the compulsory IEs caused by the UEs on the arm k itself as it is required $\hat{\iota}_{k,t} \leq \hat{\tau}_{k,t}$ in `PC-CapUL` 2.

As for the first group of IE, we have the number of these IE is less than

$$\sum_{i=1,i\neq k,\mu_i\geq\frac{1}{2}\mu_k}^{K} \frac{2304\sigma^2 m_i^2}{\mu_i^2} \log(T)$$

according to the analysis in Theorem 5. And similarly the number of the third group can be bounded by $2 \cdot \frac{1152\sigma^2 m_i^2}{\mu_i^2} \log(T)$. We can bound the number of the first and the third group of IE as:

$$\sum_{i=1,i\neq k,\mu_i\geq\frac{1}{2}\mu_k}^{K} \frac{2304\sigma^2 m_i^2}{\mu_i^2} \log(T) + \frac{2304\sigma^2 m_i^2}{\mu_i^2} \log(T)$$

$$\leq \sum_{i=1,\mu_i\geq\frac{1}{2}\mu_k}^{K} \frac{2304\sigma^2 m_i^2}{\mu_i^2} \log(T)$$

$$\leq \sum_{i=1,\mu_i\geq\frac{1}{2}\mu_k}^{K} \frac{9216\sigma^2 m_i^2}{\mu_k^2} \log(T)$$

$$\leq \frac{9216M^2\sigma^2}{\mu_k^2} \log(T)$$

As for the second group of IE, we can employ the lemma 7 to bound them:

$$\sum_{i=1,\mu_i\leq\frac{1}{2}\mu_k}^{K} \frac{32\sigma^2 \log(T)}{(\mu_i - \mu_k)^2} + 1$$

$$\leq K + \sum_{i=1,\mu_i\leq\frac{1}{2}\mu_k}^{K} \frac{128\sigma^2 \log(T)}{\mu_k^2}$$

$$\leq K + \frac{128K\sigma^2}{\mu_k^2} \log(T)$$

Then we reach the lemma that gives the upper bound of $\mathbb{E}[\hat{\tau}_{k,T}]$:

**Lemma 8.** *In our algorithm, the expected number of IE on arm k is limited with an upper bound as:*

$$\mathbb{E}[\hat{\tau}_{k,T}] \leq \frac{9216M^2\sigma^2}{\mu_k^2} \log(T) + \frac{128K\sigma^2}{\mu_k^2} \log(T) + K$$

By replacing the $\mathbb{E}[\hat{\tau}_{k,T}]$ in lemma 6 with upper bound of $\mathbb{E}[\hat{\tau}_{k,T}]$ in lemma 8, and replacing the $\mathbb{E}[\hat{\iota}_{k,T}]$ with the maximal value $\frac{1152 m_k^2}{\mu_k^2}\sigma^2\log(T)$, we get that:

$$\mathbb{E}[REG(T)]$$

$$\leq \sum_{k=1}^{K}\left(\left(\frac{9216M^2+128K}{\mu_k^2}\sigma^2\log(T)+K\right)(\mu_k-c)m_k+\frac{1152m_k^2}{\mu_k^2}\sigma^2\log(T)cN\right)$$

$$+\sum_{k=1}^{K}\left(\frac{2}{T}KT\cdot Regmax_k\right) \tag{18}$$

$$\leq \sum_{k=1}^{K}\left(\frac{9216M^2+128K}{\mu_k}\sigma^2\log(T)m_k+\frac{1152m_k^2}{\mu_k}\sigma^2\log(T)N\right)$$

$$+\sum_{k=1}^{K}(2K\cdot Regmax_k)+\sum_{k=1}^{K}(Km_k\mu_k)$$

In the second inequality we use $\mu_k > c$ for all k.

For arbitrary $\Delta$:

$$\mathbb{E}[REG(T)]$$

$$\leq \sum_{\mu_k\geq\Delta}^{K}\left(\frac{9216M^2+128K}{\mu_k}\sigma^2\log(T)m_k+\frac{1152m_k^2}{\mu_k}\sigma^2\log(T)N+K\mu_k m_k+2K\cdot Regmax_k\right)$$

$$+\sum_{\mu_k\leq\Delta}^{K}(T(\mu_k-c)m_k)$$

$$\leq \sum_{\mu_k\geq\Delta}^{K}\left(\frac{9216M^2+128K}{\Delta}\sigma^2\log(T)m_k+\frac{1152m_k^2}{\Delta}\sigma^2\log(T)N\right)+\sum_{\mu_k\leq\Delta}^{K}T\Delta m_k$$

$$+\sum_{k=1}^{K}(2K\cdot Regmax_k)+\sum_{k=1}^{K}(Km_k\mu_k)$$

$$\leq \frac{9216M^3+128KM+1152M^2N}{\Delta}\sigma^2\log(T)+TM\Delta+O(1)$$

$$=O\left(M^2\sigma\sqrt{T\log(T)}\right)$$

The last step is letting $\Delta=\sqrt{\frac{9216M^3+128KM+1152M^2N}{TM}\sigma^2\log(T)}$. ∎

In the proof of Theorem 6, we actually find a better instance-dependent regret upper bound as follows:

**Theorem 7.** *The instance-independent regret upper bound for Algorithm 2 is:*

$$\mathbb{E}[REG(T)]\leq\sum_{k=1}^{K}\left(\left(\frac{9216M^2+128K}{\mu_k^2}\sigma^2\log(T)+K\right)(\mu_k-c)m_k+\frac{1152m_k^2}{\mu_k^2}\sigma^2\log(T)cN\right)$$

$$+\sum_{k=1}^{K}2K\cdot\max(\mu_k m_k,Nc)$$

**Proof of Theorem 7.** This theorem is a direct result of the equation (18)

**Remark.** It should be noted that the regret upper bound in Theorem 5 can be very large if $\max_i \mu_i/\min_i \mu_i$ is large, and the same problem exists in Wang et al. (2022a)'s regret upper bound. The dependence of the regret upper bound on this ratio is unreasonable, and thus a better form of regret upper bound is given explicitly here.

