# OpenReview forum: "Multi-play Multi-armed Bandit Model with Scarce Sharable Arm Capacities"
_ICLR.cc/2025/Conference — Submitted to ICLR 2025_

### Official Review · Reviewer_Xyas · 2024-10-17

**Soundness:** 3
**Presentation:** 2
**Contribution:** 3
**Rating:** 3
**Confidence:** 4

**Summary:**

This paper discusses the problem of the Multi-play Multi-armed Bandit Model with Shareable Arm Capacities (MP-MAB-SAC). It tightens the lower bounds for both sample complexity and the cumulative regret compared to the previous work. Besides, this paper proposes corresponding algorithms to match the lower bounds. Finally, the numerical experiments show that the proposed algorithms outperform the existing ones.

**Strengths:**

The theoretical contributions are nontrivial. This paper shows tighter lower bounds, and then proposes new algorithms to match them. Furthermore, the experiments verified the theories.

**Weaknesses:**

I have the following concerns:

1. The writing quality of this paper falls below the standards required for publication in ICLR. Issues such as clarity, rigor, and basic grammatical correctness are prevalent. It appears that the authors did not thoroughly review the paper before submission. From a writing perspective, the paper remains in draft form: numerous typos, confusing notations, and grammatical errors hinder readability. For example,  (1) in Lemma 2, $\epsilon^{uE}$ should be $\epsilon^{UE}$; (2) in the proof of Lemma 2 “Bourel et al., 2020” is even not cited; (3) in the proof of Theorem 1, which lemma is used here? Besides, this theorem should be proved more formally; (4) What is the first baseline “MP-MAB-SA” in the experiments?

2. The explanations provided in the paper are insufficient. (1) In Section 1, more concrete examples of the model's practical applications are needed. (2) The claim that certain changes in settings make the model more suitable for LLMs requires stronger evidence. For instance, the movement cost $c$ (which is known to the learner) seems irrelevant. (3) The paper should provide a more in-depth analysis of the experimental results, going beyond mere statements of fact.

3. The comparison with the previous work seems not fair. (1) Since $N \ge M$ makes the learner only need to learn the capacity $m_k$, without needing to learn the rank of the arms, the learning task seems easier. (2) In lines 307~310, is there any evidence to show stability is getting better? Besides, I’m kind of confused about this result because the robustness v.s. regret usually has some trade-off, which means the increasing of stability may (not always) lead to the decreasing of performance.

**Questions:**

Please try to solve the problems in weaknesses. In addition, since the improvement in results achieved in this paper mainly comes from the careful selection of UCB, I would like to know what kind of inspiration it will bring to future work? This is not necessary as the theoretical improvement itself is interesting.

---

> ### Author Response · Authors · 2024-11-23
>
> Thank you for your comment, and your questions will be answered one by one.
>
> 1. As for the mentioned confusing notations in the first weakness: (1) $\epsilon^{UE}$ is the correct notation; (2) the full name of the cited article is ''Tightening exploration in upper confidence reinforcement learning'';(3) the lemma used in the proof of Theorem 1 is the Theorem 14.2 in Lattimore$\&$Szepesvari(2020). This lemma is also mentioned in the proof of Theorem 3 and it is cited properly there;(4)The mentioned "MP-MAP-SA" should be "MP-MA-SE" actually;(5)
> A rebuttal version of the article is submitted and the writing quality is improved.
>
>
> 2. The question mentioned in the second term in the weaknesses part:(1) Sorry that we are not sure whether more examples or more concrete details on the LLM application are expected. We would appreciate it if you can give a clue on this. (2) Running an LLM service has cost on computing resource, IT operations of the system, system maintenance, etc. Transmitting an end user's query to the commercial LLM server has a communication cost, especially when a query has a long prompt input. The cost $c$ is an aggregate abstraction of the above cost.  (3) More detailed analysis of the experimental results is proposed in Section 6 in the rebuttal version and Section A in the appendix.
>
> 3. The question mentioned in the third term in the weaknesses part: (1) The comparison is fair for the following three reasons. First, the comparison between the results of the example complexity is fair. The example complexity is focused on one particular arm, and the rank of arms is not included. Second, as for the regret bounds, consider the case when $M=N$, which satisfies the conditions in both Wang et al.(2022a) and this article. Then the optimal action is the same, and the rank of the arms is not required to be learned. It should be noted that the regret lower and upper bounds in Wang et al.(2022a) are reduced to $\Omega\left(\sum_{k}\log T\right)$ and $\Omega\left(\sum_{k}\left(w_k\sigma^2 m_k^2/\mu_k^2\right)\log T\right)$. Compared to the bounds in this article, their lower bound is less informative than the $\Omega\left(\sum_k\log T/\mu_{k}^2\right)$, while their upper bound is the same. Third, Wang et al calculate the regret by decomposing it into several parts. And the part that is related to estimating the capacities of the ''optimal arms'' is not related to ranking of the arms, and the part of regret related to estimating the capacities is bounded as it is shown above. So it is fair to compare the part that estimates the capacities in the regret bounds. (2) The word ''unstable'' might mislead you about the robustness of the algorithm. The actual meaning is that the denominators of the confidence bounds are preferred to be positive as soon as possible. Otherwise $N-\sum_{i=1,i\neq k}^K m_{i,t}^l$ will be the upper bound of the capacity, and this value is not a good estimation. Additional experiments are conducted to compare the convergence speeds of the two estimators
>
> 4. The question mentioned in the questions part: The article shows that the sample complexity gap is closed by UE and IE, in which the arms are assigned with UCB and LCB plays respectively. The closed sample complexity gap serves as clues that the regret bound gap in this problem can be closed by UE and IE as well.

---

> > ### Comment · Reviewer_Xyas · 2024-11-28
> >
> > Thanks for author's rebuttal. Generally speaking, my concerns still remain. Here are some follow-up comments:
> >
> > * When reading the updated version, I still find some simple mistakes in Appendix. I hope the author will thoroughly read and revise the entire paper to prevent hindrance to the reader's comprehension.
> > * The author's reference to LLM applications in Lines 96-104 lacks sufficient introduction and explanation. The author does not give reasonable argument in the subsequent content, which makes me feel that the author is just trying to introduce some unrelated but popular topics.
> > * I appreciate the author's explanation of experiments. Could you please provide an error bar to support the statistical significance of the results?
> >
> > Considering the limitations of the current version, I've decided to keep my score unchanged.

---

> > > ### Author Response · Authors · 2024-11-28
> > >
> > > Thank you for your further comment, and your questions will be answered one by one.
> > >
> > > 1. An improved version of this article has been submitted already.
> > >
> > > 2. We make a careful balance between generality and customization in developing the model.  As for the customization, our model aims to capture the core resources allocation challenge in the scarce resources scenarios, and our model can be applied to LLM, edge computing, cognitive ratio applications, online advertisement placement etc. As for the generality, our model serves as a backbone, which could be further adapted to specific applications with the reward function extended.
> > >
> > > 3. There is not enough time for additional experiments, considering the time your reply was sent to us. Your requirement of error bars in the figures is reasonable and they will be added in the future version.

---

### Official Review · Reviewer_HpxS · 2024-10-31

**Soundness:** 3
**Presentation:** 3
**Contribution:** 2
**Rating:** 5
**Confidence:** 3

**Summary:**

This paper considers the problem of multi-play multi-armed bandits with scarce shareable arm capacities. Specifically, different from [Wang et al., 2022a], this paper considers the problem where $N\geq \sum_k m_k$ where $m_k$ is the capacity of action $k$. With a modification on the reward function, this paper proposes new sample complexity lower/upper bound that is tight as well as regret lower/upper bound for this problem. Specifically, the author claims that the sample complexity lower bound proven in this paper improves upon the one shown in [Wang et al., 2022a]. Empirical results are also shown to strengthen their theoretical findings.

**Strengths:**

- This paper first considers this problem with scarce shareable arm capacities and proposes both lower and upper bound for both sample complexity and the regret bound.
- Based on the parts that I checked, the proofs look correct to me.
- The experiments are also conducted to show superior performance compared to the previous work.

**Weaknesses:**

- One main concern is the motivation of this paper to consider the case where $N\geq \sum m_k$. In this case, the problem seems to be easier (in the sense of algorithm design) since you will definitely explore each action sufficiently enough to figure out the exact $m_k$ while in the opposite case $N< \sum m_k$, the problem seems to be harder since you need to decide the exploration amount for the suboptimal $k$. Can the authors explicitly justify the choice of studying the $N\geq \sum m_k$ case and why it is challenging compared to the previous case?
- This also leads to the question about the comparison between the lower/upper bound shown in this paper and [Wang et al., 2022a]. While the authors claim better lower bound, I wonder whether the upper/lower bound are comparable in these two cases? Can the algorithm that is derived in this setting adapted to the other? Moreover, I am not sure why equation (5) is more reasonable since it makes sense to me to have the noise's variance larger when $m_k$ or $a_k$ is large.
- As for the upper bound, the bounds in Theorem 5 seems to be suboptimal since it seems to be dependent on $\frac{\max_i \mu_i}{\min_i \mu_i}$, which can be large.
- I do not understand the lower bound argument shown in Theorem 4. When the cost $c=0$, then this ratio becomes 0, which is surely not informative. In addition, why is the ratio independent of $m_k$? Can the authors explain more on this?
- Typos:
  - Line 223: it -> if
  - Line 224: a_k -> a_{t,k}?
  - Line 471: depended -> dependent
  - Line 751: missing lemma reference.
  - missing periods at the end of many theorem statements (e.g. Theorem 4,5,6..)

**Questions:**

See "Weakness" Section for questions.

- I wonder whether the dependency on the cost parameter $c$ can be improved for the regret lower bound.

**Details Of Ethics Concerns:**

None.

---

> ### Author Response · Authors · 2024-11-23
>
> Thank you for your comments, and your questions will be answered one by one.
>
> 1. First, the choice of studying the $N\geq \sum m_k$ case is mainly attributed to the fact that it is common in real world that the resources are scarce and the competition for these resources is intense. Second, it is not mentioned in the article that the $N\geq \sum m_k$ case is more challenging than the $N< \sum m_k$ case. You might be curious about the fairness of the comparison between results in both cases, and this question will be answered below.
>
> 2. (1) The comparison is fair for following three reasons. First, the comparison between the results of the example complexity is fair. The example complexity is focused on one particular arm, and the rank of arms is not included. Second, as for the regret bounds, consider the case when $M=N$, which satisfies the conditions in both Wang et al.(2022a) and this article. Then the optimal action is the same, and the rank of the arms is not required to be learned. It should be noted that the regret lower and upper bounds in Wang et al.(2022a) are reduced to $\Omega\left(\sum_{k}\log T\right)$ and $\Omega\left(\sum_{k}\left(w_k\sigma^2 m_k^2/\mu_k^2\right)\log T\right)$. Compared to the bounds in this article, their lower bound is less informative than the $\Omega\left(\sum_k\log T/\mu_{k}^2\right)$, while their upper bound is the same. Third, Wang et al calculate the regret by decomposing it into several parts. And the part that is related to estimating the capacities of the ``optimal arms'' is not related to ranking of the arms, and the part of regret related to estimating the capacities is bounded as it is shown above. So it is fair to compare the part that estimates the capacities in the regret bounds.(2) As for the equation 5, the returned reward of equation 5 contains less information as it is explained in line 79-85. And it is a reward model of conventional linear bandits with one dimensional feature, which is as common as the one with increased noise's variance. Moreover, we can get the corresponding regret lower bound with reward function as equation 1 in the same way, and the lower bound is modified by multiplying the term $\exp({-9TK/2})$. This implies that the problem setting with equation (5) is more difficult than that with equation 1. Consequently, if we can get a closed regret upper and lower bounds in the setting of equation (5), the same techniques or insights might work in the setting of equation 1.
>
> 3. That's correct. This bound is also a kind of rough bound, and we list it as a theorem for readers to find the similarity of this the upper bound and the one in Wang et al.(2022a). And your concern is solved in the rebuttal version. A new bound is shown explicitly as Theorem 7 at the last part of the appendix.
>
> 4. When the movement cost $c=0$, consider the case when there is almost infinite plays compared with the capacities. We can assign every arm with one play in the first round, and double the number of plays assigned to each arm after every round. Because of the absence of movement cost, once the number of plays assigned to arm $k$ exceeds the capacity $m_k$, there would be no regret. So the regret of this strategy remain constant after only a few rounds. This implies that there might be other strategies which optimize the action in much fewer rounds without learning the exact $m_k$. As for the next question, the results in sample complexity in Theorem 1 and Theorem 2 show that the sample complexity is independent of $m_k$. And in the proof of the lower bound, we can only assume that the regret generated by single exploration to be $\min(\mu_k-c,c)$. So it is reasonable for the regret lower bound to be independent of $m_k$.

---

> > ### Comment · Reviewer_HpxS · 2024-11-29
> >
> > I thank the reviewers for their efforts. With the new theorem 7, the authors improve upon the upper bound rate. However, I still have questions about my other concerns.
> >
> > 1. The reason that the regret can be constant when c=0 (in the lower bound construction) still does not make sense to me since when you double the number of play, you may violate the condition that \sum a_k \leq N. From another perspective, the proven regret upper bound does have addition leading terms when c=0, showing that the lower bound can be loose. I also feel that m_k should appear in the lower bound and the current lower bound result that is m_k-free seems to ignore this m_k constraint.
> >
> > 2. I also do not understand why the lower bound in Wang et al., 2022 applies here since the reward model (the noise scale) is different. A more rigorous comparison is needed I think. In addition, a smaller proven lower bound does not show that one problem is easier since a trivial regret lower bound for all problems is 0. A smaller upper bound instead shows this.

---

> > > ### Author Response · Authors · 2024-11-30
> > >
> > > Thanks a lot for your further comment, and your questions will be answered one by one.
> > >
> > > 1. You might be confused by the difference between the sample complexity and regret.  Sample complexity measures the number of explorations that a strategy requires to estimate the capacity correctly, and it is proven to be $m_k$-free here. It is mentioned in this article that the sample complexity gap is closed, and the sample complexity lower bound is tight. However, the regret lower bound is not asserted to be tight in this article. A closed gap in the regret upper and lower bounds is one of the targets of our future work. It is not sure whether the regret can be bounded by a constant if there is no movement cost. The optimal action is not unique when $c=0$, and $a\_t $ with $a_{k,t}\geq m_k$ are all optimal actions. The mentioned strategy in our last reply serves as an illustration that some strategies can find the optimal action set without learning the exact $m_k$. Though the doubling strategy is only valid for large $N$( $N=O(T^{100}2^T)$ for instance ), some clues imply the existence of efficient strategies even in conventional settings.
> > >
> > > 2. There are mainly two reasons of why the lower bound of Wang et al.(2022a) applies here. (1) Comparison between the lower bound in this article and the one in Wang et al.(2022a) is based on the comparison between the information contained in the reward functions. With less informative reward function, a more informative regret lower bound is proposed in this article, implying that the lower bound of Wang et al.(2022a) is kind of loose. Considering the case when $\mu=0,c=0$ on an arm, learning the capacity $m_k$ is impossible in our setting, but possible in Wang et al.(2022a)'s setting by checking the variance. (2) The lower bounds derived with information theoretical methods are usually the estimators of the problem difficulty, and in both this article and Wang et al.(2022a) this opinion is admitted. One can check that with the same information theoretical methods used in this article to get the complexity lower bound, he or she may get a complexity lower bound modified by multiplying a term of $O(\exp(-T))$ in Wang et al.(2022a)'s setting.

---

### Official Review · Reviewer_fizY · 2024-11-04

**Soundness:** 4
**Presentation:** 3
**Contribution:** 3
**Rating:** 8
**Confidence:** 4

**Summary:**

This paper studies the multi play multi-armed bandit problem having shared arm capacity where the in each round, the learner gets to select the arm for a number of pulls capped by the capacity limit with the goal of maximizing the total reward at the end of the play. The authors propose a new reward function and develop a new algorithm PC-CapUL for this problem setting. The developed algorithm provides tighter bounds on sample complexity and regret in comparison to the existing works, efficiently balances exploration and exploitation. The work is applicable in resource allocation problem with capacity constraint scenarios such as LLM inference and many other real world scenarios.

**Strengths:**

•	The problem of Multi play multi-armed bandit problem is an interesting setting to study and improve the foundation of it as it pertains to main real-world settings including LLM inference serving. The work re-establishes that with emphasis on theoretical guarantees.

•	The work provides theoretical improvements in sample complexity compared to the existing work on  MP-MAB-SAC. It tends to close the sample complexity gap found in the previous work in Reference A

•	The authors also provide a new Improved algorithm, PC-CapUL that performs much better than other existing algorithms and have a solid theoretical backing to it with proved theoretical Regret bound guarantees.

•	The experiments cover the regimes where the number of arms is larger which predominantly requires more exploration to take place. The developed algorithm provides much better performance in terms of regret compared to other existing algorithms in this experimental setting.

Reference:
 [A] Xuchuang Wang, Hong Xie, and John C. S. Lui. Multiple-play stochastic bandits with shareable finite-capacity arms. International Conference on Machine Learning, ICML 2022.

**Weaknesses:**

•	The experimentation design could have been done much better with the inclusion of better baseline comparison in addition to the algorithm found in Reference A . Also, utilizing a real-world dataset for evaluation would have further complemented these theoretical results.

•	The readability of the paper could be much improved. Also, a brief intuitive explanation like a proof sketch could be added in the main text to help the reader get the intuitive logic and understanding of the proof techniques.

•	A more detailed theoretical comparative analysis like how regret fares against the regret of other algorithms would make the argument much stronger for the developed PC-CapUL algorithm. Moreover, having such a discussion would also help us uncover insights like how the regret bound behaves in different regimes.

Reference:
 [A] Xuchuang Wang, Hong Xie, and John C. S. Lui. Multiple-play stochastic bandits with shareable finite-capacity arms. International Conference on Machine Learning, ICML 2022.

**Questions:**

1.	$m_k$ is deterministic and well known beforehand as to how many pulls can be made in a round. However, there is a constant movement cost $c$ associated to an arm. In case of LLM query, the number of pulls is associated to the amount of query a server instance can handle.

Are $m_k$ and moving cost $c$ dependent in this scenario? If so, how do this implication sit with all the theoretical proof, or do they have to be independent? A clarification on this would help the readers utilize the developed algorithms on many scenarios where the dependencies are crucial.

2.	Why is ordering of plays of arm selection important in $a_t$, providing some details on it will avoid ambiguity around its objective of whether to maximize the resource utilization or to maximum capacity of the arm?

3.	Also, with respect to movement cost, in the experiment setting, it has been assigned to an arbitrary value of 0.1 .  Is there any fundamental reason for that? Also, how can they be evaluated in a practical scenario when they are also coupled with reward formulation? Adding some details around them can greatly improve the clarity of the work.

4.	It would be nice to see how the experiments scale up with varying parameters like changing the $m_k$ and changing the movement cost etc ? This will help us understand the empirical performance of the algorithm much better.

Reference:
 [A] Xuchuang Wang, Hong Xie, and John C. S. Lui. Multiple-play stochastic bandits with shareable finite-capacity arms. International Conference on Machine Learning, ICML 2022.

---

> ### Author Response · Authors · 2024-11-23
>
> Thank you for your comments and suggestions, and your questions will be answered one by one.
>
>
> 1. The $m_k$ is not known in advance. There is no requirement on the dependence of $m_k$ and $c$ in this article. The only requirement of the movement cost $c$ is that $c<\mu_k$ for all $k$, so that every unit of occupied resource is expected to increase rather than reduce the utility. This requirement is necessary in our proof, especially when considering the regret lower bound. Because it is demanded that both overestimating and underestimating the capacity $m_k$ should generate a regret.
>
>
> 2. The main reason is that there are not sufficient plays for every arm to be played with UE freely. Once there are enough plays to arrange UE on all arms, we can use Theorem 2 on each arm separately. When there are not enough plays, some arm are forced to be played with IE. According to the regret lower bound in Theorem 3 and Theorem 4, the larger $\mu_k$ is, the less time slots we have to spend on that arm to get $m_k$. Moreover, if the arm with large $\mu_k$ is forced to be played with IE because of the lack of plays, it will be likely that the regret can be large too ( we do not know $m_k$ in advance, so all we can do is using $\hat{\mu}\_{k,t}$ or  $\hat{\mu}\_{k,t}m\_{k,t}^u$ or other estimators to predict the regret). In this algorithm and the proof of its regret upper bound, it is shown that $\hat{\mu}\_{k,t}$ is a qualified estimator.
>
> 3. The only reason for setting $c=0.1$ is to meet the requirement that $c<\mu_k$ for all $k$. The movement cost $c$ is known in advance in a practical scenario, and $c$ is independent of $\mu_k$. Additional experiments with different values of $c$ are conducted, and results are proposed in Appendix A.4.
>
> 4. There are changes in the range of $m_k$ in the appendix part A.1. And it is shown that the larger $m_k$'s range is, the larger the regret is. Additional experiments are done as you required, and you can check the figures depicting the impact of changing $c$ in the Appendix A.

---

> > ### Comment · Reviewer_fizY · 2024-11-27
> >
> > Thank you for the response. I went through all the answers and reviews. I have no further questions at this point. However, to improve clarity and to avoid ambiguity regarding the ordering of arm selection, the authors could incorporate the relevant information from the Q2 comments into their next version of this paper.

---

> > > ### Author Response · Authors · 2024-11-28
> > >
> > > Thanks a lot for your further suggestion.
> > >
> > > Additional explanation about the ordering of arm selection has been added in the front of the proof of the regret upper bound (Theorem 6) in the appendix part. Since the rank plays a significant role in the proof of subsequent Lemma 7 and Theorem 6, with further explanation here, it will be easier for readers to comprehend why the arms should be ranked according to $\hat{\mu}\_{k,t}$ and why this ranking can improve the regret upper bound of the algorithm.

---

### Official Review · Reviewer_P6Qi · 2024-11-05

**Soundness:** 3
**Presentation:** 1
**Contribution:** 3
**Rating:** 6
**Confidence:** 3

**Summary:**

This paper revisits multi-play multi-armed bandit with shareable arm capacities problem. Improved on previous work Wang et al. (2022a), the paper proposes refined lower and upper bounds for both sample complexity and regret. For sample complexity, the authors propose a minmax lower bound, and give an algorithm that matches the bound. For regret, the authors provide both instance dependent and instance independent regret lower bounds, and find algorithms that match the bounds up to some model-dependent factors.

**Strengths:**

1. The work closes the sample complexity gap and narrow the regret gap for the MP-MAB problem. Although the techniques used in the proof are not particularly unique (mostly based on regular UCB and LCB), the conclusions are still very interesting and make sense.
2. The work propose numerical simulation to show the advantages of their algorithms.

**Weaknesses:**

1. The writing is a bit poor. The paper contains many colloquial expressions, i.e., line 383 "But if", line 390, 403, 405 "And furthermore" "And this".
2. The author states in the introduction that the algorithm has applications to LLM inference serving. I believe it’s necessary to provide some LLM-related experiments to support this statement.

**Questions:**

1. should we assume that $\mu_k\ge c$ for all $k$? The authors state that the optimal action is always $(m_1,\dots, m_K)$ in line 211. It seems that this only holds when $\mu_k\ge c$ for all $k$.
2. what is \mu in Theorem 1. I did not find the definition.
3. I am curious about how could the sample complexity in Theorem 2 gets rid of the dependence of N. Intuitively, even there is no noise (sigma = 0), for any algorithm, it still need at least $\log N$ rounds to find the true $m_k$ by binary search. Is the dependence on $N$ hidden in $\xi$?


Typos:
line 224: $a_{k,t}$ instead of $a_k$
line 289: for large probability -> with high probability
line 383: to played with -> to play with

---

> ### Author Response · Authors · 2024-11-23
>
> Thank you for your comment, and your questions will be answered one by one.
>
> 1. For the first question, the answer is yes. If for some $k$ there is $\mu_k<c$, then the arm $k$ can not generate reward. In most real world cases, it is rare to see people paying to occupy resources and get nothing. So we can set the movement cost as a small value.
>
> 2. That is a mistake in notation. $\mu_k$ is set as $\mu$ here, which is the per unit reward of the arm $k$.
>
> 3. Sorry that I do not understand the question because there should be no dependence of the play number $N$ in the sample complexity, but I think you may be curious about the absence of $m_k$ in the upper bound of the sample complexity. Sample complexity here is the number of time slots demanded to calculate the true capacity $m_k$. The estimated UCB is not searched from $[N]$ randomly, but calculated with previous information generated by previous actions in the MAB. If the confidence intervals in Wang et al. (2022a) are used in this article, there will always be $m_k$ in the upper bound of the sample complexity. But with the refined confidence intervals in this article, the gap between upper and lower bounds is closed up. The sample complexity is different from the regret bound, and the regret upper bound depends on $N$. As for the cases when there is no noise, the upper bound of the sample complexity is actually $2$: one UE and one IE should suffice to get the correct capacity via dividing. So the complexity for all non-negative $\sigma$ can be modified by adding $2$, and this new upper bound is shown in the rebuttal version.

---

### Meta-Review · Area_Chair_fdFv · 2024-12-22

**Metareview:**

This paper addresses the multi-play multi-armed bandit with shareable capacities problem, presenting results on improved sample complexity, regret lower bounds, algorithms, and regret upper bounds. The primary concern with this paper lies in the subtle differences between the scenarios it addresses and those in prior work, raising questions about the fairness and validity of comparisons with existing results and lower bounds.

Specifically, the paper focuses on cases where the number of plays $N$ exceeds the total amount of capacities $M$. However, this restriction might simplify the problem, and the paper does not provide a convincing explanation to justify this aspect. Additionally, there are several areas where the clarity and rigor of the writing, both in terms of narrative and mathematical descriptions, are lacking.

For these reasons, I cannot support the acceptance of this paper at this time.

**Additional Comments On Reviewer Discussion:**

The reviewers raised concerns regarding the subtle differences between the scenarios addressed in this paper and those in prior work, questioning whether comparisons with existing results and lower bounds are fair and justifiable. Specifically, the paper focuses on cases where the number of plays $N$ exceeds the total amount of capacities $M$, but this restriction might simplify the problem, and the paper fails to provide a convincing explanation to address this issue.

Additionally, the reviewers noted that many parts of the paper lack clarity and rigor, both in terms of textual expression and mathematical descriptions. The authors' rebuttal did not sufficiently alleviate these concerns.

---

### Decision · Program_Chairs · 2025-01-22

Reject